# Effective Intervention Features of a Doping Prevention Program for Athletes: A Systematic Review with Meta-Analysis

**DOI:** 10.3390/sports13040108

**Published:** 2025-04-07

**Authors:** Luis Felipe Reynoso-Sánchez, Amairani Molgado-Sifuentes, Hussein Muñoz-Helú, Jeanette M. López-Walle, Diego Soto-García

**Affiliations:** 1Research Centre for Physical Culture Sciences and Health, Autonomous University of Occident, Culiacán 80014, Mexico; amairani.molgado@uadeo.mx; 2Department of Economic-Administrative Sciences, Autonomous University of Occident, Los Mochis 81217, Mexico; hussein.munoz@uadeo.mx; 3Faculty of Sport Organization, Autonomous University of Nuevo Leon, San Nicolas de los Garza 66455, Mexico; jeanette.lopezwl@uanl.edu.mx; 4Department Physical and Sport Education and Group Research AMRED, University of Leon, 24007 Leon, Spain; dsotg@unileon.es

**Keywords:** anti-doping education, intervention program, cognitive-based education, values-based education, athletes, anti-doping knowledge, doping attitudes, doping susceptibility

## Abstract

This study systematically reviewed the effectiveness of cognitive, affective, and combined approaches in doping prevention, considering the impact of athletes’ active versus passive participation. Following the PRISMA 2020 guidelines and the PICOS framework, a literature search identified ten studies involving 3581 athletes (1094 women, 2487 men). Ten studies were included as a sample in the meta-analysis and meta-regression, which were used in the effect size calculation. This meta-analysis shows that anti-doping education programs effectively improve short-term doping intentions (ES = 0.29, *p* < 0.001) and anti-doping behaviors (ES = −0.27, *p* < 0.001), although there is a decline in the long-term effects (ES = −0.13, *p* = 0.009). Moral behaviors were unaffected (ES = 0.01, *p* < 0.001), suggesting that changing deeper values requires alternative approaches like mentorship. Passive participation negatively impacted doping intentions (ES = −0.40, *p* = 0.004) and behaviors (ES = −0.40, *p* = 0.022), highlighting the need for active engagement. Pre-experimental designs showed a negative effect on behaviors (ES = −0.74, *p* = 0.023), emphasizing the importance of rigorous methodologies. While anti-doping education programs effectively influence short-term attitudes and intentions, sustaining behavioral change requires continuous reinforcement and active engagement. The decline in the long-term effects suggests that standalone interventions are insufficient to instill lasting anti-doping behaviors in athletes.

## 1. Introduction

Doping in sports remains a significant global challenge, with implications not only for the physical and mental health of athletes, but also for the fairness and integrity of sports competitions. The prevention and eradication of doping among athletes at all levels of competition has been a long-standing goal pursued by the World Anti-Doping Agency (WADA), whose rules are set out in the World Anti-Doping Code [1]. Prohibited substances and methods used to enhance performance have been the subject of extensive research, and international sports organizations, including the International Olympic Committee (IOC) and WADA itself, have developed rigorous regulations to combat this issue. However, for more than a decade, WADA has acknowledged that doping prevention must go beyond sanctions to include educational and psychological interventions that address the underlying factors that drive athletes to resort to doping. This understanding led to the establishment of the WADA International Standard for Education (ISE) for anti-doping education programs in 2019 [2], providing guidelines for designing educational interventions. This standard emphasizes that an athlete’s first exposure to doping-related issues should come through an educational program rather than a doping test or, even worse, direct contact with banned substances.

In this context, the evolution of doping prevention approaches has integrated not only informational and regulatory strategies, but also psychosocial approaches that explore the role of athletes’ beliefs, emotions, and attitudes in decision making [3,4]. According to WADA’s ISE [2], among these approaches are cognitive (information programs that aim to provide anti-doping education) and affective (education programs that aim to transfer anti-doping knowledge with an emphasis on personal values and principles development) models, which, based on social learning [5] and/or the health belief model [6,7], as well as socio-affective theories [8,9], respectively, offer complementary perspectives on how athletes can be guided and encouraged to strengthen their sense of self-efficacy to avoid the use of prohibited substances. Additionally, within preventive interventions, the importance of the roles assumed by athletes during these programs has also been considered, whether through active participation, where they are directly involved in the process, or passive participation, where they simply receive information without deeper engagement [10].

### 1.1. Cognitive Approach to Doping Prevention

The cognitive approach to doping prevention is based on the premise that athletes’ beliefs, knowledge, and perceptions about doping play a decisive role in their behavior. This approach is grounded in cognitive learning theories, which hold that individuals process and store the information they receive, something that later influences their decisions and actions [11]. In the context of doping, this implies that an athlete who has adequate knowledge of the risks associated with the use of prohibited substances (in terms of health, as well as the legal consequences) is more likely to avoid their use [12].

One of the primary goals of the cognitive approach is to provide athletes with accurate, scientifically based information on the adverse effects of doping and the benefits of legal and safe alternatives for performance enhancement. Through educational programs and informational campaigns, athletes can learn about the types of prohibited substances, their harmful effects in the short and long term, and the sports and legal sanctions they will face if such substances are detected [11]. Additionally, this approach seeks to develop critical thinking skills in athletes, training them to properly assess risky situations and make informed decisions.

From a theoretical standpoint, the cognitive approach is based on models such as those related to social learning theories, which consider the educational process as a behavioral model based on the behavior of others, specially, relevant persons for the learners [13], and the health beliefs model, which makes the individual’s decision to engage in health-related behaviors dependent on their perceived susceptibility to harm, the severity of the consequences, and the perceived benefits and barriers to taking preventive action [6]. This approach focuses the anti-doping interventions on the capabilities of the people delivering the program in terms of incorporating the relevant information about doping and anti-doping behavior, contextualizing real cases as examples of doping risks and putting a major part of the effort into increasing their knowledge about anti-doping tools in order to reduce the susceptibility of the participants to doping [14].

In the context of doping, this implies that if athletes have negative perceptions toward the use of prohibited substances, they also have the perception that their environment also rejects doping, and they believe that they have enough control to abstain from the use of such substances, thus they are less likely to engage in these practices [15]. However, although the cognitive approach is effective in increasing knowledge and awareness about the risks of doping, it has certain limitations. Information alone is not always sufficient to change behavior, as athletes may be influenced by other factors, such as competitive pressure, the perception of short-term benefits from doping, or a lack of emotional support [16]. This is where the affective approach plays a crucial role, addressing the emotional and motivational dimensions that also influence the decision to resort to doping.

### 1.2. Affective Approach to Doping Prevention

The affective approach focuses on the role that the emotions, motivations, and attitudes of the athlete play in doping prevention. Unlike the cognitive approach, which primarily centers on the transmission of information and the development of cognitive skills, the affective approach explores how the athlete’s emotional experiences and personal identity influence their decision making [17]. This approach is based on psychological theories that posit that emotions and motivation have a significant impact on human behavior and that addressing these aspects can be key to preventing doping [18,19], and is also based on psychosocial theories, such the Theory of Planned Behavior [20], which posits that an individual’s intentions to carry out a behavior are influenced by their attitudes, subjective norms, and the perception of their control over that behavior.

One of the main objectives of the affective approach is to foster an ethical identity and emotional commitment to the values of clean sport. This involves working with athletes to develop a positive attitude toward fair competition and a sense of personal pride in their performance without resorting to illegal aids [21,22]. Through affective interventions, such as emotional coaching, the promotion of self-reflection, and the development of self-esteem, the goal is to strengthen athletes’ emotional resilience in the face of competitive pressure and the temptation to engage in doping [23].

Along the same lines, another explanatory theory is the Self-Determination Theory (SDT) [24], which posits that human motivation can be intrinsic (driven by enjoyment or personal satisfaction) or extrinsic (driven by external rewards). In the context of doping, athletes who are intrinsically motivated by a love of the sport and personal growth are less likely to resort to doping compared to those who are mainly motivated by extrinsic factors, such as achieving titles or social recognition [18,25]. Affective interventions specifically aim to strengthen intrinsic motivations, encouraging athletes to value their achievements based on their effort and skills rather than on dishonestly obtained results.

Like the cognitive approach, the affective approach also has its limitations. Emotions, though powerful, can be difficult to control, especially in high-pressure situations, such as elite sports competitions. Additionally, some athletes may be resistant to exploring or changing their emotions due to cultural or personal barriers. Therefore, integrating both cognitive and affective approaches can provide a more comprehensive and balanced approach to doping prevention.

### 1.3. Active and Passive Participation in Doping Prevention Interventions

In addition to cognitive and affective approaches, another relevant aspect of the effectiveness of preventive interventions is the level of athlete participation in programs designed to combat doping. Active and passive participation by athletes in these programs can significantly influence intervention outcomes, as the way athletes engage in the prevention process affects their understanding and adoption of strategies.

According to the literature [10,26], active participation is characterized by the athlete’s direct and proactive involvement in prevention activities. This can include participation in interactive workshops, ethical debates, doping simulations, and collaboration in designing solutions to maintain a doping-free sports culture. From a theoretical perspective, active participation aligns with active learning, which suggests that individuals assimilate and retain information more effectively when they are directly involved in the educational process [27,28]. Moreover, SDT has demonstrated that involving athletes in decision making and the learning process can enhance their satisfaction in terms of their autonomy and competence needs [18]. This reinforces the benefits of adopting an active role for participants in anti-doping education programs [26]. Therefore, it can be concluded that by actively engaging, athletes not only receive information, but they also apply it, reinforcing their sense of personal responsibility and their ability to make informed decisions.

Conversely, passive participation implies that athletes act as recipients of information, attending preventive programs with limited interaction or reflection. While this type of intervention can increase the participant’s basic knowledge about doping, research suggests it is less effective in producing long-term behavioral changes [12,29]. Athletes who participate passively may internalize some concepts; however, according to SDT, the satisfaction of the participant’s basic psychological needs is negatively impacted (particularly autonomy and competence) by educational or training activities that do not provide individuals with opportunities to engage in decision making and to actively contribute to their learning process [30,31]. This lack of engagement not only diminishes the participant’s motivation, but also restricts critical thinking and limits the practical application of acquired knowledge, ultimately reducing the effectiveness of interventions [18]. Combining active and passive participation within a preventive program can offer a balanced approach that addresses the diverse needs and learning styles of athletes. While some athletes may benefit from a more structured and receptive approach, others may need more dynamic engagement to better integrate learned concepts and apply them to their daily lives in sport [32].

Since WADA and researchers have focused their attention on ways to prevent doping in regard to competitive, amateur, and non-competitive athletes, many kinds of interventions have been developed, addressing the need to provide information about doping and the relevance of prevention for health and/or fair play [11,12,33]; nonetheless, previous systematic research and meta-analyses are limited by the scarcity documents available about educational intervention programs [6,26,34] or they point out all the literature related to anti-doping knowledge building, including observational and experimental data [4]. Recently, an expanded meta-analysis carried out by Ntoumanis et al. [35] has provided valuable information regarding the impact of interventions and experimental research on the doping intentions and attitudes of participants, presenting partial conclusions about the effect of intervention programs on this variable, and suggesting the need to explore the impact on other variables, such moral disengagement and social norms, as key factors to prevent doping behaviors.

In an attempt to aid the purpose of WADA’s ISE for the development of educational anti-doping programs that help to increase positive behaviors against doping, such as increasing anti-doping knowledge and morals, aimed at maintaining fairness in sport, the objective of this systematic literature review was to assess the effectiveness of cognitive and affective approaches, as well as the impact of athletes’ participation, in enhancing the outcomes of doping prevention programs, since the International Standard for Education was proposed by WADA in 2019.

## 2. Materials and Methods

### 2.1. Search Strategy

A systematic literature review was conducted using the Preferred Reporting Items for Systematic Reviews and Meta-analyses (PRISMA) methodology [36] (Appendix A) and was registered on the International Prospective Register of Systematic Reviews (PROSPERO ID: CRD420251020655). Studies were included if the interventions evaluated improvements in knowledge and behaviors following the implementation of educational programs and/or outreach on the consumption of doping substances among athletes from 2019 to 4 October 2024. The literature search was conducted in two different phases. The first exploration took place from 15 to 20 April 2024, during which two researchers conducted searches in four databases: PubMed, ScienceDirect, SciELO, and Redalyc. Subsequently, the search was refreshed on 4 October 2024, using the same databases.

#### 2.1.1. Eligibility Criteria

The definition of the eligibility criteria was based on the PICOS approach, which specifies the participants, interventions, comparisons, outcomes, and types of studies to be considered in advance [37]. (P) Population: Studies involving athletes of any level, sport, gender, age, and nationality. Studies conducted on students who were not athletes or on coaches without an evaluation of their athletes were excluded, as the aim was to understand the effect of anti-doping education on athletes. (I) Intervention: All interventions for doping prevention were considered, including audiovisual materials, forums, presentations, educational games, surveys, and group or individual educational sessions, with pre- and post-intervention assessments to evaluate the knowledge gained, as well as the effect on the athletes’ anti-doping behavior. Studies that did not assess the learning effect were excluded. (C) Comparison: All the intervention groups (i.e., anti-doping education program, anti-doping moral intervention, e-learning anti-doping program), control groups, or population groups were considered for the comparison. (O) Outcome: The studies must evaluate the effect of the intervention with at least one post-test measurement of moral disengagement, intention to dope, doping behavior, doping likelihood, doping susceptibility, anti-doping knowledge, or self-efficacy. (S) Study design: Experimental, clinical trials or randomized clinical trials within an educational intervention or educational stimuli were included.

#### 2.1.2. Search Terms

The search strategy contained a combination of “Medical Subject Headings” (MeSH) and free words for related key concepts, including the following: (i) PubMed: (“doping prevention” [Title/Abstract] AND “antidoping program” [Title/Abstract]) OR “antidoping education” [Title/Abstract] OR (“Antidoping” [All Fields] AND “promotion” [Title/Abstract]) OR (“Antidoping” [All Fields] AND “learning” [Title/Abstract]) OR “education to legality” [Title/Abstract] OR (“Antidoping” [All Fields] AND “guideline” [Title/Abstract]) AND “doping intention” [Title/Abstract]) OR “doping behavior” [Title/Abstract] OR “attitudes toward doping” [Title/Abstract] OR “doping susceptibility” [Title/Abstract] OR “doping likelihood” [Title/Abstract] OR “antidoping knowledge” [Title/Abstract] + Filters: Years “2019–2024”. (ii) ScienceDirect: (Doping prevention) AND (Antidoping program OR antidoping education OR antidoping learning OR antidoping guideline) AND (Doping intention OR Doping behavior OR attitudes toward doping OR doping susceptibility OR Doping likelihood) + Filters: Years “2019–2024”; article type: research articles; subject areas: medicine and dentistry/psychology/social sciences; languages: English. (iii) Redalyc: (Athletes) AND (Doping prevention) AND (Antidoping program OR antidoping education OR antidoping learning OR antidoping guideline) AND (Doping intention OR Doping behavior OR attitudes toward doping OR doping susceptibility OR doping likelihood) + Filters: Years “2019–2024”; languages: English/Spanish; discipline: health/psychology/medicine/multidisciplinary (social science)/education. (iv) SciELO: ((Doping prevention) OR (Antidoping program) OR (antidoping education) OR (antidoping learning) OR (antidoping guideline) OR (Doping intention) OR (Doping behavior) OR (attitudes toward doping) OR (doping susceptibility) OR (doping likelihood)) + Filters: Years “2019–2024”; WoS thematic areas: sport/education/humanities/medicine/psychology.

### 2.2. Selection Criteria

#### 2.2.1. Inclusion Criteria

To be included in this review, the studies had to meet the following criteria: (i) implement an intervention with the aim of promoting anti-doping education and/or reducing doping-related behaviors; (ii) athletes as the study participants; (iii) at least one pre/post-intervention measurement; (iv) primary or secondary outcomes related to doping or anti-doping knowledge, attitudes toward doping, doping behavior, doping intention, doping likelihood, doping susceptibility, moral identity and moral disengagement in regard to doping, anticipated guilt in regard to doping, and self-regulatory efficacy to resist doping; (v) involve clinical trials, randomized clinical trials, experimental, or quasi-experimental research; and (vi) published in Spanish or English.

#### 2.2.2. Exclusion Criteria

Studies that contain any one of the following exclusion criteria were not considered in this review: (i) cross-sectional or experimental study designs without an educational intervention program; (ii) reviews, meta-analyses, or nonoriginal studies; (iii) studies that involved only coaches or students (non-athlete population); (iv) studies published before 2019; and (v) articles wherein the full text could not be retrieved.

### 2.3. Extraction and Synthesis of the Data

A data extraction checklist was created for each study included in the review. The following details were collected: (i) the last name of the first author and publication year; (ii) study aim; (iii) the country where the study was conducted and the sample characteristics; (iv) the study design and measurement scales used; (v) a description of the intervention; and (vi) the relevant results. Two researchers conducted the data extraction, which involved an independent screening using Covidence.

### 2.4. Statistical Analysis

The primary outcome of this study was the effect of the intervention on doping behavior, positive anti-doping factors, and anti-doping moral factors. To assess these effects, the effect size (ES) was calculated as the mean difference [38], evaluating changes in performance based on the intervention approach and the athlete’s role in the educational program, both pre-test and post-test. When applicable, changes between the pre-test and follow-up test were also analyzed.

Two studies [39,40], wherein the results were reported as proportions, data transformation was performed using the Campbell Effect Size Calculator for ordinal data to obtain the ES and standard error of the differences.

The overall ES calculations were conducted using JASP (version 0.19.3.0), applying a random effects model (Restricted Maximum Likelihood). The confidence intervals were set at 95%. Study heterogeneity was assessed using Cochran’s Q test (*p* < 0.05), while inconsistency was measured using the *I*^2^ statistic [38].

Whenever possible, the data were coded for different subgroups to examine the influence of potential moderator variables, such as study design, type of intervention (cognitive vs. values based), the athlete’s role during the interventions (active vs. passive), the age of the participants, and the total number of sessions in the intervention program. Exploratory analyses were carried out, using meta-regression for continuous variables and subgroup analysis for categorical variables, both performed using JASP. The significance level was set at 0.05.

### 2.5. Assessment of Methodological Quality

The methodological evaluation of the selected studies was conducted using the Joanna Briggs Institute (JBI) assessment tools for the risk of bias for randomized controlled trials [41] and quasi-experimental studies [42] in order to exclude research with inadequate methodologies.

## 3. Results

### 3.1. Selection of Studies

A search across four databases using the specified criteria yielded a total of 180 articles: ScienceDirect (*n* = 7), PubMed (*n* = 66), SciELO (*n* = 5), and Redalyc (*n* = 102). Duplicate articles were removed using Mendeley^®^ and the automation tool Covidence^®^, resulting in 179 articles. Then, two independent researchers (LFR-S and AMS) reviewed the article titles to select those meeting the inclusion criteria. Any disagreements were resolved using Covidence^®^, leading to only three discrepancies, which were addressed through the use of a third review (HM-H). This process resulted in the inclusion of 11 articles. Finally, the abstracts of these 11 articles were reviewed, and only those involving interventions concerning athletes were selected, resulting in a final set of nine articles. Following the full-text review process, the authors identified in the references one study that had not been found during the systematic review [16] that met the inclusion criteria for the review, bringing the total number of articles included in the final analysis to 10 studies (Figure 1).

### 3.2. Methodological Quality

All the randomized clinical trial (RCT) studies included (five) in the review met the minimum methodological quality standards, achieving a score of 10 or higher (see Table 1) and all the quasi-experimental studies included (five) in the review obtained 7 points or more (see Table 2), which is considered good.

It can be observed that most of the cluster randomized trial (CRT) studies reduced the bias in terms of the internal validity (items 1–3), while none of the studies met the requirement of full blinding for the intervention and outcome assessment due to the specific nature of the intervention examined. Finally, two RCT studies did not report that they had taken intention-to-treat (ITT) analysis into account in their results (item 11). As for the quasi-experimental studies, three out of five studies had no control group, and only one study showed a lack of evidence of reliability in the way the outcomes were measured.

### 3.3. Characteristics of Participants and Study Design

Table 3 shows the characteristics of the athlete participants. A total of 3670 athletes were involved in the studies (1094 women and 2487 men), with a minimum age of 14 years. Also, 130 coaches (33 women and 95 men) took part in one study reviewed [43], and 32 non-athletes (16 women and 16 men) were involved in another study [48]. Two of the studies were conducted with an all-male sample [39,40], and one study omitted the gender characteristics of a portion of the sample [43]. With respect to the sport level classification, three studies did not report the specifications of the sporting level of their athlete sample [16,23,43]. The rest of the studies had mixed participation involving national or international level athletes, which can be considered a high level of performance, as well as regional and local level athletes.

Five documents reported the study design as a cluster randomized trial (CRT), with at least two groups and two measures [16,23,43,44,45]. Two of these CRT studies did not employ a control group to compare their results [16,23], while another study used the WADA National Anti-Doping Organizations’ (NADOs) standard anti-doping education program as a control group to compare the results [43]. The other five studies included in the present review were based on quasi-experimental designs to analyze the effects of the intervention programs [39,40,46,47,48]. Only two of these quasi-experimental studies reported the inclusion of control groups to compare the results [46,48].

All of the studies reviewed included the measurement of the changes in the outcomes as a consequence of the anti-doping intervention program. The most repeated variables measured were doping intention/doping likelihood (six studies) [16,23,43,44,46,48], moral disengagement [16,23,43,44,46], and anti-doping knowledge [40,43,46,47,48], with five studies each, attitudes toward doping (four studies) [39,43,45,48], and three studies each for doping susceptibility [44,45,47], anticipated guilt [16,23,44], and anti-doping self-efficacy [16,43,46], and two for anti-doping practices [43,47]. Also, measures focused on evaluating the internal and external moral values and their relationship to doping behaviors were evaluated in three studies [23,44,47], as well as one study involving coaches, which reported on the influence of basic psychological needs support, needs-related thwarting and satisfaction, and coaches ability to transmit anti-doping knowledge on doping behaviors to athletes [43].

### 3.4. Evaluation of the Intervention Programs

According to the aim of our research, the analysis of the intervention programs was structured in regard to the domain (cognitive/affective) and characteristics of the intervention, as well as the role of the athlete during the program (passive/active). A cognitive domain-based intervention was observed in all the reviewed studies, being used alone in eight studies [16,23,39,40,43,44,47,48] and combined with the affective domain in four of the them [43,44,45,46], while only the affective domain was applied in both studies carried out by Kavussanu et al. [16,23].

Four of the reviewed studies conducted a single intervention session, two of them lasting between 60 and 80 min [47,48], while the procedures conducted by da Silva et al. [40] and Thomas et al. [39] used reduced time periods, such as 6 min of a video game and the review and explanation of an anti-doping information poster. Only one piece of research [45] managed two 90 min intervention sessions for two of their study groups, as well as providing access to an online platform (the time was not taken into account, but rather the fulfilment of the platform’s objectives) for the third intervention group. Another study handled four sessions of 90 min each for their intervention program [46], while three other investigations applied six sessions for their anti-doping program, with a duration of 45 min [44] and 60 min [16,23]. The research by Ntoumanis et al. [43] reported two different durations for the intervention program, applying 12 sessions of 60 min each for the intervention group, while a single 60 min session was designated for the control group.

### 3.5. Meta-Analysis of the Studies

In total, two overall effect sizes were calculated for each outcome variable (doping intention, anti-doping behaviors, and anti-doping moral factors), assessing the change from pre-test to post-test and from pre-test to follow-up, yielding a total of six overall ESs (Figure 2, Figure 3 and Figure 4 for the pre-test–post-test ES; Appendix A for the pre-test–follow-up ES). The global ES for doping intentions showed a significant effect for the pre- to post-test comparison (*ES* = 0.32; *CI* 95% = 0.25, 0.39; *N* = 26; *I*^2^ = 45.79; *Q* = 30.10; *p* = 0.037), as well as for the pre-test to follow-up comparison (*ES* = 0.34; *CI* 95% = 0.26, 0.42; *N* = 19; *I*^2^ = 31.63; *Q* = 20.55; *p* = 0.114). Regarding anti-doping behaviors, a significant ES was also obtained for the pre- to post-test comparison (*ES* = −0.29; *CI* 95% = −0.51, −0.08; *N* = 16; *I*^2^ = 94.37; *Q* = 96.17; *p* < 0.001); nevertheless, the ES for anti-doping behavior maintenance was not significant when the pre-test result was compared with the follow-up (*ES* = −0.13; *CI* 95% = −0.29, 0.03; *N* = 8; *I*^2^ = 73.29; *Q* = 15.32; *p* = 0.004). The last ES calculated the anti-doping moral behaviors, which showed no significant effect neither for the pre- to post-test comparison (*ES* = 0.01; *CI* 95% = −0.10, 0.12; *N* = 32; *I*^2^ = 77.26; *Q* = 108.80; *p* < 0.001) nor for the pre-test to follow-up comparison (*ES* = −0.01; *CI* 95% = −0.14, 0.14; *N* = 30; *I*^2^ = 0.00; *Q* = 3.39; *p* = 1.00).

The moderation effects were calculated for the characteristics of the intervention design to determine the influence of the type of educational program (cognitive vs. affective), the athlete’s role (active vs. passive), age, intervention length (0 to 1 sessions = short; 2 to 4 sessions = intermediate; 5 or more sessions = long), and for the pre- to post-test analysis; the method design (CRT, quasi-experimental or pre-experimental) was included as a moderator. The meta-regression effects (see Table 4) for the pre- to post-test analysis indicates the athlete’s role as significant in regard to doping intention (*F* = 34.52; *df* = 1.18; *p* = <0.001), as well as study design (*F* = 12.73; *df* = 2.18; *p* < 0.001), age (*F* = 17.49; *df* = 1.18; *p* < 0.001), and number of intervention sessions (*F* = 16.60; *df* = 2.18; *p* = <0.001), whereas for anti-doping behavior, the athlete’s role (*F* = 8.22; *df* = 1.9; *p* = 0.019) was also significant. No significant moderators were obtained for anti-doping moral behavior.

With respect to the meta-regression for the pre-test to follow-up comparison, the athlete’s role (*F* = 24.62; *df* = 1.14; *p* = < 0.001), age (*F* = 14.43; *df* = 1.14; *p* = 0.002), and number of intervention sessions (*F* = 9.23; *df* = 2.14; *p* = 0.003) was shown to be a significative moderator of the ES of doping intention. No other significant moderator was observed for the pre-test to follow-up analysis of anti-doping behavior, as well as for any analysis of anti-doping moral behavior.

### 3.6. Descriptive Evaluation of the Results

#### 3.6.1. Knowledge on Doping

Five of the studies reviewed [40,43,46,47,48] examined changes in doping knowledge in a total of 965 IG and 638 CG participants. Two of these studies [46,48] used WADA-developed questionnaires to assess anti-doping knowledge, while two others [43,47] used self-developed and validated questionnaires, and the study by da Silva et al. [40] assessed anti-doping knowledge with a self-developed questionnaire that measured athletes’ knowledge of positive and negative substances and supplements. In terms of outcomes, two of the studies that reported on pre–post intervention anti-doping knowledge without CG participants [40,47] showed a significant increase in doping knowledge, as well as two other studies that observed a post-intervention increase in anti-doping knowledge for IG participants compared to CG participants [46,48]. One of the five studies [43] found no differences between the groups and the time.

#### 3.6.2. Doping Intention

Six of the ten studies reviewed [16,23,43,44,46,48] assessed changes in doping intention or likelihood of doping involving a total of 1400 GI athletes and 505 CG athletes. Four studies [16,23,44,47] applied the doping likelihood scale, based on two adapted hypothetical scenarios reported in previous research [49,50], while the other two used the willingness to take prohibited substances scale [43] and the scale of doping likelihood related to the benefits and costs of doping [48]. Only one study out of six [44] reported no change in doping intention after the intervention program. Similarly, in the study by Deng et al. [48], no changes were observed in the post-intervention e-learning anti-doping education program for IG participants, but differences were found between athletes and non-athletes for this variable. Another study [16] observed a lower level of doping likelihood after a psychological-based intervention, whereas this was not the case for the educational intervention.

#### 3.6.3. Doping Susceptibility

Only three of the total articles reviewed measured the doping susceptibility variable [44,45,47] in a total of 1317 GI and 387 CG participants. The studies by Hurts et al. [47] and Manges et al. [44] used a single item to measure doping susceptibility, while Nicholls et al. [45] applied the doping susceptibility subscale with five items from the Adolescent Sport Doping Inventory (ASDI, [51]). Two studies [45,47] found changes in athletes’ doping susceptibility after the intervention, with a significant decrease in perceived susceptibility in regard to the use of doping substances compared to the CG participants and pre-intervention. The third study that included this variable reported no significant changes after the intervention or in comparison to the CG participants [44].

Similarly, one study [47] included the intention to use dietary supplements to understand how this variable can be related to doping susceptibility. A single item to evaluate the extent to which athletes’ agreed to use dietary supplements in the next months was used in the study involving 302 athletes. After the intervention, the results revealed a decrease in the intention to use dietary supplements. Also, this study revealed that moral values moderate the indirect effect of the anti-doping program in reducing doping susceptibility via a reduction in the scores related to the intention to use dietary supplements.

#### 3.6.4. Attitudes Toward Doping

Of the total studies reviewed, four measured changes in the attitudes towards doping [39,43,45,48] involving a total of 1401 IG and 508 CG participants. None of the investigations used the same scale to measure this variable; the Performance Enhancement Attitude Scale (PEAS) was applied by Deng et al. [48], while Thomas et al. [39] applied an adapted version for bodybuilding athletes. Nicholls et al. [45] used the doping attitudes subscale by the ASDI, whilst Ntoumanis et al. [43] used a scale proposed by Barkoukis et al. [25] involving four items. Two intervention programs comparing IG versus CG participants found a significant change in doping attitudes for the IG group [45,48], while another study without CG participants reported a decrease in positive IG doping attitudes after the intervention [39]. The latter study observed no significant changes in this variable for the IG participants [43].

#### 3.6.5. Moral Disengagement

As for the variable moral disengagement, it was assessed in five of the reviewed articles [16,23,43,44,46] involving a total of 1380 GI and 647 CG athletes. To measure this variable, four studies [16,23,43,44] used the Moral Disengagement Doping Scale [49], while the other piece of research [46] used a similar scale, which was adapted from different previous studies. Four studies found a reduction in moral disengagement values after the intervention program [16,23,44,46]; only one study reported no changes [43]. Also, Manges et al. [44] reported a significant decrease for IG participants compared to CG participants, while Kavussanu et al. [16] observed a greater reduction for the IG psychology-based intervention compared to the IG education-based intervention.

#### 3.6.6. Self-Regulatory Efficacy to Resist Doping and Anti-Doping Practices

The analysis of changes in self-efficacy to resist doping was measured in 719 GI and 574 CG subjects in three studies [16,43,46]. Two of the investigations [16,46] used the same scale with seven items to assess the variable, and the other study used a similar scale with six items to identify how able the athlete was to resist the use of doping substances. After the intervention program, two studies reported a significant increase in the observed variable compared to the baseline [16,46]; in addition, Kavussanu et al. [16] found a greater increase in the education-based intervention compared to the psychology-based intervention.

On the other hand, anti-doping practice was evaluated by two studies [43,47], with a sample of 746 IG and 462 CG participants. Both studies used a similar tool to assess the athletes’ likelihood of engaging in anti-doping practices, such as checking sport supplements to ensure they are free from prohibited substances. After the intervention, only the study by Hurts et al. [47] observed a significant increase in terms of this variable.

#### 3.6.7. Psychosocial Doping-Related Variables

Moral values and the spirit of sport were measured in 302 athletes in one study [47] to determine the primary values related to fair play and respect for sport. Similarly, other research assessed the moral identity of 303 athletes [23] to determine changes in their moral traits after the intervention program. In terms of the results, Hurts et al. [47] observed an increase in moral values and the spirit of sport at the post-intervention time, while the other study [23] found no significant differences between the moral-based and education-based intervention program, as well as when comparing the pre-, post-intervention, and follow-up measures in both groups.

Collective moral norms and the moral atmosphere were assessed by Manges et al. [44] and Kavussanu et al. [23], respectively. The Collective Moral Attitude in Sport Groups scale [52] was applied to 248 IG and 73 CG athletes [44], while a self-developed six-item scale was used in another study to assess 303 athletes, divided into two different intervention groups [23]. Both studies found no significant changes in these variables after the application of the intervention program.

Whistleblowing [47] and empathy [44] were two variables reported by two different studies, with a sample of 302 and 248 IG and 73 CG athletes, respectively. Whistleblowing was assessed using four statements related to the participant’s agreement in terms of their likelihood to report doping by other athletes, while empathy was assessed using a subscale of a personality questionnaire. The results of the study by Hurts et al. [47] showed an increase in the likelihood of reporting doping after the intervention; Manges et al. [44] also observed an increase in this regard during the follow-up after the values-based intervention program.

## 4. Discussion

The findings of this meta-analysis provide valuable insights into the effectiveness of anti-doping education programs, particularly in relation to cognitive and affective approaches, as well as the role of athlete engagement in the learning process. The analysis of the ES revealed significant short-term improvements in doping intentions and anti-doping behaviors, although the long-term impact varied across different outcomes. Additionally, the moderation analysis highlighted the crucial role of athlete engagement, with passive participation negatively influencing doping-related attitudes and behaviors. Additionally, other variables, such as study design, participant age, and intervention length (number of sessions), were found to influence athletes’ doping intentions and anti-doping behaviors.

The global effect size for doping intentions indicated a significant positive shift from pre-test to post-test (*ES* = 0.32; *CI* 95% = 0.25, 0.39), with an even greater effect observed from the pre-test to the follow-up (*ES* = 0.34; *CI* 95% = 0.26, 0.42). These findings suggest that anti-doping education programs are effective in fostering short-term changes in athletes’ attitudes toward doping, and, in some cases, these changes persist over time. However, the effectiveness of these interventions in influencing actual anti-doping behaviors was less pronounced. Although a significant change in anti-doping behavior was observed immediately post-intervention (*ES* = −0.29; *CI* 95% = −0.51, −0.08), this effect was not sustained at the follow-up (*ES* = −0.13; *CI* 95% = −0.29, 0.03). This decline in behavioral impact over time aligns with previous research, which indicates that while educational interventions can shift attitudes, maintaining behavioral change requires continuous reinforcement and support [10,11,53].

One of the most notable findings was the lack of significant effects on anti-doping moral behaviors, both in the short term (*ES* = 0.01; *CI* 95% = −0.10, 0.12) and long term (*ES* = −0.01; *CI* 95% = −0.14, 0.14). This suggests that while cognitive and affective approaches can influence knowledge and intentions, they may not be sufficient to alter deeper moral convictions related to doping. Moral development is a complex process influenced by social, cultural, and personal factors, which might explain why educational interventions alone did not significantly impact this domain [21,50,54]. Future research should explore alternative strategies, such as values-based education and long-term mentorship, to strengthen the moral reasoning aspect of anti-doping education. Furthermore, researchers must center their efforts on team culture, identifying the group’s socio-personal traits and moral identity, including not only athletes, but also coaches and other staff members and parents.

Educational intervention programs for athletes aim to reduce doping intentions and promote anti-doping behaviors. According to Social Learning Theory, athletes are more likely to adopt anti-doping behaviors if they observe and imitate role models who promote clean sports [14,55]. However, if their post-intervention environment normalizes doping, these behaviors may fade over time. The result of our meta-analysis indicates that while changes in doping intentions tend to persist in the long term, improvements in anti-doping behaviors are often short lived and diminish over time. This difference can be attributed to several factors. Intentions are primarily shaped by internalized attitudes and values, which, once established, may require less external reinforcement [34]. In contrast, behaviors depend on external factors, such as social norms, competitive pressure, and ongoing support. Without continuous reinforcement or an environment that actively promotes anti-doping conduct, athletes may struggle to maintain these behaviors over time [7].

Moreover, many interventions focus on short-term attitude changes, but lack strategies for sustaining behavioral change [53]. The Theory of Planned Behavior explains how an intervention can positively influence attitudes, social norms, and perceived control to shape doping intentions, but if athletes feel they lack control over their environment, their behaviors may not be sustained. The absence of practical components, personalized follow-ups, or adaptive strategies can weaken the long-term impact of anti-doping behaviors [53,56]. To enhance the effectiveness of these programs, it is essential to integrate values-based education, create a supportive sports culture, and implement continuous monitoring and reinforcement mechanisms [2,12,29]. By addressing both cognitive and behavioral aspects, intervention programs can foster lasting changes in terms of both doping intentions and anti-doping behaviors [14,25,26].

The moderation analysis provided further insights into the factors influencing the effectiveness of these programs. The athlete’s role emerged as a significant moderator, with passive participation negatively impacting the doping intention (*ES* = −0.83; *p* < 0.001) and anti-doping behavior (*ES* = −0.77; *p* = 0.019) intervention effect in the pre- to post-test analysis. Additionally, passive engagement continued to be a significant moderator of doping intentions at follow-up (*ES* = −0.86; *p* < 0.001). These results align with Self-Determination Theory [57], which posits that active engagement in learning enhances autonomy, competence, and fosters long-term intrinsic motivation. When interventions support these needs, athletes internalize anti-doping values, reducing doping intentions over time. However, if their post-intervention environment fails to reinforce these psychological needs, anti-doping behaviors are less likely to endure [10,19,30,32]. To ensure lasting effectiveness, interventions must extend beyond education and create a supportive, reinforcing environment.

When athletes passively receive information without being involved in decision-making or interactive activities, they may internalize less of the educational content, reducing its long-term impact. As evidenced in the meta-analysis, the role of the athlete during the intervention programs was a significant factor in terms of their efficacy. It is described as the type of athlete participation; if the athlete interacts with the process and progress of the sessions, works with role models, or constructs materials that reflect the interpretive experience of the participants, they were considered to have an active role. Conversely, if the athlete’s participation was limited to paying attention to information, answering questions, or completing an evaluation of the intervention program without further analysis, they were considered to have a passive role. The literature supports the promotion of active roles of participants in educational intervention programs aimed at health promotion, such as doping prevention. Active involvement improves empowerment and adherence to healthy behaviors by allowing participants to take a leading role in managing their health by developing practical competencies and skills [58], significantly reducing the risk of engaging in risky behaviors [59].

The results of the literature review show that out of a total of 13 groups of athletes who participated in intervention programs, nine adopted an active role, while four adopted a passive role. The study by Ntoumanis et al. [43] indirectly measured the effect of the intervention on the athletes because it focused the program on the coaches, who redirected what they learned to their athletes. According to the meta-regression analysis, a passive role in intervention programs had a negative moderating effect on doping intentions and anti-doping behaviors. These findings align with the existing literature, which emphasizes the importance of active participation in health promotion programs, including anti-doping education [27]. For instance, using hypothetical scenarios where athletes engage in group discussions, role playing, and problem-solving exercises have been shown to encourage active participation [23,44]. Other effective strategies include brainstorming sessions, selecting values for specific situations, and explaining their reasoning in response to questions or dilemmas [16,45]. Additionally, intervention programs incorporating digital tools, such as video games [40,46] or web-based platforms [45], have been used to guide athletes through decision-making processes related to doping prevention, further fostering engagement and active learning.

Similarly, the design of the intervention played a role in shaping the outcomes. In the pre- to post-test analysis, a quasi-experimental design was associated with a significant negative impact on doping intention (*ES* = −0.76; *p* < 0.001). This suggests that more rigorous experimental methodologies may be better suited to capturing the true effects of educational programs. The lack of significant moderators for the follow-up analysis of anti-doping behavior and anti-doping moral behaviors suggests that additional factors, such as social influences or environmental reinforcement, may play a stronger role in the long-term adherence to anti-doping principles [21,60].

On the other hand, age was found to be a moderator of changes in doping intentions following intervention programs (*ES* = 0.12; *p* < 0.001) and even at follow-up (*ES* = 0.13; *p* = 0.002). The findings indicate that age serves as a moderator in the effectiveness of educational intervention programs aimed at reducing doping intentions among athletes. A previous meta-analysis [35] aimed at evaluating the effect of experimental designs on doping intentions did not find any moderating influence in regard to the age of the participants. Our study confirms that, specifically, older athletes exhibit greater reductions in doping intentions following such interventions, both immediately after the program and at follow-up assessments. This trend may be attributed to older and more experienced athletes’ increased awareness of the adverse consequences associated with doping, leading to more negative attitudes toward its use [61,62]. Furthermore, age-related factors, such as greater maturity and a heightened sense of responsibility, may contribute to older athletes’ responsiveness to anti-doping education. These factors can enhance their ability to internalize the information provided during interventions, leading to more significant changes in their intentions regarding doping.

Finally, the last tested moderator was the intervention length (the number of sessions in the intervention program). The findings indicate that longer intervention programs (six or more sessions) have a negative impact on reducing doping intentions both immediately after the program (*ES* = 0.12; *p* < 0.001) and at follow-up (*ES* = 0.13; *p* = 0.002). This counterintuitive result may be attributed to several factors, such diminishing returns, where in regard to longer programs additional sessions contribute less to behavioral change. This phenomenon has been observed in various behavioral change interventions, where the initial sessions have the most significant impact, and subsequent sessions yield progressively smaller benefits [63]. In addition, longer interventions may lead to participant fatigue, reduced engagement, or increased dropout rates, impacting negatively the overall efficacy of the program [64]. Another explanation for this result is the lack of immediate application of new knowledge or skills in real-world settings [65]. Immediate application is crucial for reinforcing behavioral change, and prolonged programs might postpone this reinforcement, leading to reduced effectiveness.

All the studies analyzed for this systematic review focused on promoting anti-doping program interventions aimed at influencing athletes’ knowledge, behaviors, and morals in relation to doping. Many methodological differences can be observed in the included studies, as five employed a CRT, but only two of them [44,45] used a passive CG (without intervention), the rest of them compared IG with CG participants that received the standard anti-doping education program promoted by WADA’s NADOs (active CG). The other five studies based their research on a quasi-experimental design, but only two of them [46,48] compared the results of IG participants with passive CG participants, while the other three studies had no CG to compare. According to the literature, there are advantages and disadvantages related to the use of a passive or active CG in research intervention programs, but a recent meta-analysis examining cognitive interventions [27] reveals that the type of control group does not significantly affect the effect sizes for objective cognitive measures. Whereas a passive CG allows changes in the experimental group relative to the “natural” baseline to be observed, an active CG enables a comparison to be carried out as to whether an innovative treatment is more effective than the existing treatment. In this sense, the analysis of the results in the studies reviewed should be carried out considering the specific methodological characteristics of each study.

### 4.1. Practical Implications

From a practical standpoint, these findings have important implications for the design and implementation of anti-doping education programs. The evidence suggests that programs should prioritize active learning approaches, where athletes are encouraged to participate in discussions, decision-making exercises, and the real-life application of anti-doping principles. Interactive methods, such as scenario-based learning and peer-led interventions, may enhance engagement and improve the retention of key messages [34]. Additionally, interventions should extend beyond a one-time educational session and incorporate long-term reinforcement strategies, such as periodic refresher courses, mentorship, and integration with broader ethical education frameworks.

Another consideration is the integration of cognitive and affective elements within anti-doping education. While this meta-analysis did not identify significant moderation effects based on the type of educational program (cognitive vs. affective), previous literature suggests that a combination of both approaches is likely to be most effective [2,4,35,44]. Cognitive approaches enhance knowledge and critical thinking, while affective strategies target emotions and moral engagement, fostering a more holistic understanding of the consequences of doping.

In summary, future anti-doping educational interventions must consider the following:*Mixed cognitive and affective approaches to interventions;**The promotion of active and collaborative learning processes;**The provision of multiple sessions, but the avoidance of very long intervention programs;**The use of mentoring figures to reinforce values, morals, and anti-doping behaviors;**An emphasis on motivational enhancement to avoid relapses in doping behaviors after the program concludes.*

### 4.2. Limitations and Future Research Directions

This systematic review has several limitations, primarily stemming from the methodological inconsistencies across the studies analyzed. The first possible bias is related to the fact that the search was limited to only free-access platforms, leaving out potential articles indexed in prestigious databases such as SPORTDiscus or Web of Science, which may have limited the scope of the analysis. Additionally, restricting the selection to published documents in peer-reviewed journals excluded other unpublished manuscripts, such as WADA’s Social Science Research Reports or theses. However, this decision was made to ensure the highest possible quality of the included studies.

Another limitation of this analysis is the cut-off criteria used to evaluate the period since the introduction of WADA’s ISE in 2019. Some studies that demonstrated the effectiveness of educational interventions before the publication of WADA’s ISE may have been excluded from the research. Nonetheless, advancements in technology and pedagogy within anti-doping education have influenced how interventions are designed and implemented. Including only recent studies allows us to capture contemporary approaches that integrate new methodologies, such as interactive digital platforms and personalized learning strategies [66].

Furthermore, variations in control group (CG) types, such as the use of passive CGs (without intervention) versus active CGs (receiving standard anti-doping education), pose challenges in comparing the intervention effects consistently. Moreover, the inclusion of both controlled randomized trials (CRTs) and quasi-experimental designs, where some studies lack a control group altogether, limits the ability to assess the true effectiveness of the interventions. The diversity in the study design, combined with smaller sample sizes and different intervention focuses (cognitive, affective, or combined), also restricts the generalizability of these findings across broader athlete populations. The above limitation was previously emphasized, but it is difficult to expand the scope of an intervention due to the lack of solid structures, economic resources, interest, and capacity of the different organizations involved to develop it, and due to the novelty of this problem in society [6,67].

Future research in regard to anti-doping educational intervention programs should strive for greater methodological standardization in evaluating anti-doping interventions. According to Petróczi et al. [68], a total solution to the doping phenomenon is extremely complex, but it can be mitigated. Thus, the use of as many tools and approaches as possible is needed to have a better impact on the results of doping prevention in athletes, which is a critical element for anti-doping education programs. Employing CRTs with both passive and active CGs could allow a more nuanced understanding of intervention effectiveness, while larger sample sizes and more diverse participant demographics [67] would improve the generalizability of the results. In the present review, more than 50% of the educational programs were applied in two European countries (United Kingdom and Greece), while only just above 25% of the interventions were carried out in Asia, America, and Oceania countries, which highlights the need to broaden the horizons of exploration of this phenomenon in regard to other cultures and contexts. Additionally, integrating longitudinal study designs would help researchers assess the long-term impact of these interventions on athletes’ doping-related knowledge, attitudes, and behaviors. Given the observed benefits of combining cognitive and affective approaches, future studies might also explore the optimal balance between these methods to enhance anti-doping program efficacy [44]. Lastly, incorporating more active roles for participants, as suggested by the findings, could further strengthen interventional outcomes by empowering athletes to take a proactive stance against doping.

## 5. Conclusions

In conclusion, this meta-analysis highlights the effectiveness of anti-doping education programs in influencing short-term doping intentions and behaviors, although long-term maintenance remains a challenge. The role of active athlete engagement emerged as a crucial factor, emphasizing the need for interactive and participatory learning methods. Moving forward, anti-doping education should incorporate strategies to sustain behavioral change over time, such as continuous reinforcement, mentorship, and integration with broader ethical and psychological frameworks. By addressing these factors, future interventions can maximize their impact on promoting clean sport and ethical decision making among athletes.

The findings suggest that cognitive-based programs are effective in increasing athletes’ anti-doping behaviors, such as their knowledge and self-efficacy, as well as affective-based programs. A combination of these approaches appears to offer the most comprehensive benefits, as it addresses both the athletes’ cognitive understanding and intrinsic motivation. Additionally, promoting an active role for athletes within these programs enhances engagement and empowers them to internalize anti-doping principles, potentially leading to more sustained behavioral change.

However, methodological inconsistencies, such as varying control group types and study design, highlight the need for greater standardization in future research. Standardizing these elements would facilitate more direct comparisons across studies and provide clearer insights into intervention efficacy. As anti-doping education continues to evolve, future research should focus on refining these interventions through longitudinal and larger-scale studies to ensure that anti-doping efforts effectively shape athletes’ long-term attitudes and behaviors, ultimately fostering a lasting culture of integrity and fair play in sports.

## Figures and Tables

**Figure 1 sports-13-00108-f001:**
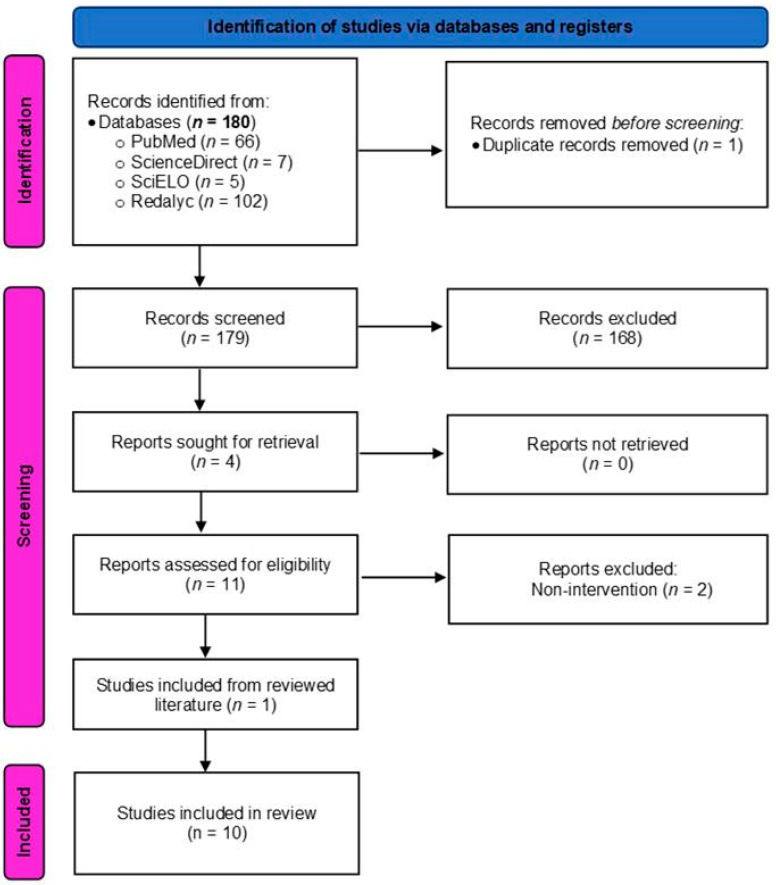
PRISMA flow chart and diagram detailing the process of article identification, screening, and assessment for final inclusion in the review.

**Figure 2 sports-13-00108-f002:**
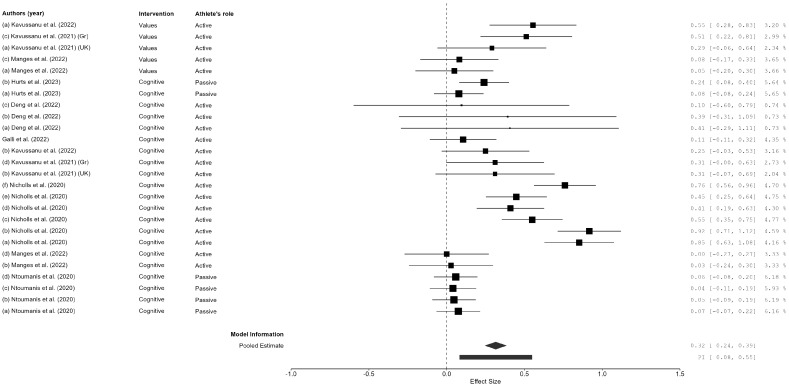
Forest plot of the pre–post differences in the effect of the anti-doping intervention program on doping intention. Figure references [16,23,43,44,45,46,47,48].

**Figure 3 sports-13-00108-f003:**
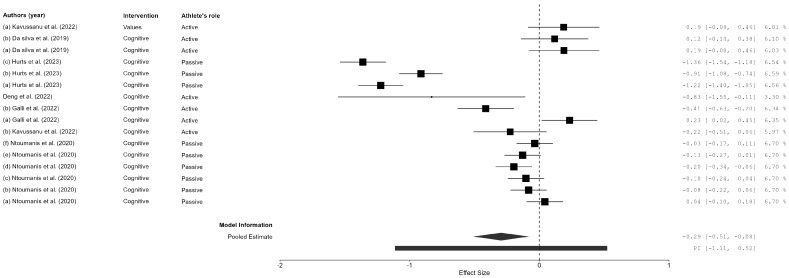
Forest plot of the pre–post differences in the effect of the anti-doping intervention program on anti-doping behaviors. Figure references [16,40,43,46,47,48].

**Figure 4 sports-13-00108-f004:**
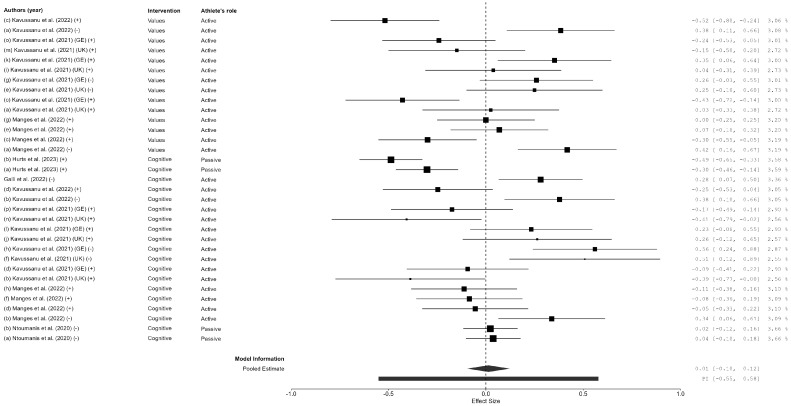
Forest plot of the pre–post differences in the effect of the anti-doping intervention program on anti-doping moral behaviors. Figure reference [16,23,43,44,46,47].

**Table 1 sports-13-00108-t001:** Results of methodological quality assessment of the included studies using the JBI Critical Appraisal Tool for the Assessment of Risk of Bias for Randomized Controlled Trials.

Study	Item													Total
	1	2	3	4	5	6	7	8	9	10	11	12	13	
Ntoumanis et al. [43]	Yes	Yes	Yes	No	No	Yes	Yes	Yes	Yes	Yes	Yes	Yes	Yes	11
Manges et al. [44]	Yes	Unclear	Yes	Yes	No	Yes	No	Yes	Yes	Yes	No	Yes	Yes	10.5
Nicholls et al. [45]	Yes	Yes	No	Yes	No	Yes	Yes	Yes	Yes	Yes	No	Yes	Yes	10
Kavussanu et al. [23]	Yes	Yes	Yes	Yes	No	Yes	No	Yes	Yes	Yes	Yes	Yes	Yes	11
Kavussanu et al. [16]	Yes	Yes	Yes	Yes	No	Yes	No	Yes	Yes	Yes	Yes	Yes	Yes	11

Note. JBI Critical Appraisal Tool for the Assessment of Risk of Bias for Randomized Controlled Trials items: (1) Was true randomization used for the assignment of participants to the treatment groups?; (2) Was allocation to the treatment groups concealed?; (3) Were the treatment groups similar at the baseline?; (4) Were the participants blind to treatment assignment?; (5) Were those delivering the treatment blind to treatment assignment?; (6) Were the treatment groups treated identically other than the intervention of interest?; (7) Were the outcome assessors blind to treatment assignment?; (8) Were the outcomes measured in the same way for the treatment groups?; (9) Were the outcomes measured in a reliable way?; (10) Was follow-up completed and if not, were differences between the groups in terms of their follow-up adequately described and analyzed?; (11) Were participants analyzed in the groups to which they were randomized?; (12) Was appropriate statistical analysis used?; (13) Was the trial design appropriate and any deviations from the standard RCT design (individual randomization, parallel groups) accounted for in the conduct and analysis of the trial?

**Table 2 sports-13-00108-t002:** Results of methodological quality assessment of included studies using the JBI Critical Appraisal Tool Checklist for Quasi-Experimental Studies.

Study	Item									Total
	1	2	3	4	5	6	7	8	9	
Galli et al. [46]	Yes	Yes	Yes	Yes	Yes	Yes	Yes	No	Yes	8
Hurst et al. [47]	Yes	No	Yes	Yes	Yes	Yes	Yes	Yes	Yes	8
Thomas et al. [39]	Yes	No	Yes	Yes	Yes	Yes	Yes	Yes	Yes	8
da Silva et al. [40]	Yes	No	Yes	Yes	Yes	Yes	No	Yes	Yes	7
Deng et al. [48]	Yes	Yes	Yes	Yes	Yes	Yes	Yes	No	Yes	8

Note. JBI Critical Appraisal Tool Checklist for Quasi-Experimental Studies: (1) Is it clear in the study what is the “cause” and what is the “effect”?; (2) Was there a control group?; (3) Were the participants included in any comparisons similar?; (4) Were the participants included in any comparisons receiving similar treatment/care, other than the exposure or intervention of interest?; (5) Were there multiple measurements of the outcome, both pre and post the intervention/exposure?; (6) Were the outcomes of the participants included in any comparisons measured in the same way?; (7) Were the outcomes measured in a reliable way?; (8) Was follow-up completed and if not, were the differences between the groups in terms of their follow-up adequately described and analyzed?; (9) Was appropriate statistical analysis used?

**Table 3 sports-13-00108-t003:** Summary of studies included in the systematic review, athlete participants, and intervention characteristics.

Reference	Population	Method	Intervention	Results
Ntoumanis et al. [43], United Kingdom (UK), Greece, and Australia	Athletes IG *n* = 444 (38.7% ♀, 43.2% ♂, 18.1% omitted). Age (M ±SD): 22.45 ±11.40 y. Athletes CG *n* = 462 (34.8% ♀, 63.6% ♂, 0.2% other, 1.4% omitted). Age (M ±SD): 18.62 ±7.07 y. Sport level: Not specified.	Study design: Cluster randomized controlled trial design with parallel group, two-condition, superiority trial. Three measures: baseline (pre-intervention), end of intervention (12 weeks), and follow-up (2 months post-intervention). Measures: 1. Willingness to take prohibited substances; 2. Doping moral disengagement; 3. Attitudes toward doping; 4. Efficacy to resist doping-related temptations; 5. Behaviors against inadvertent doping; 6. Anti-doping knowledge. 7. Perceived coach need-related support; 8. Perceived coach need-related thwarting; 9. Basic psychological needs-related satisfaction; 10. Basic psychological needs frustration; 11. Use of prohibited substances.	Name of the intervention (IG): Motivational enrichment anti-doping education. Domain: Cognitive and affective. Duration: Twelve 60 min sessions (one per week). Characteristics: Standard anti-doping education enrichment with motivational content about the supportive communication style of coaches. 1. Standard anti-doping education. 2. Introduction to the need for supportive communication. 3. How to apply the need for supportive communication to discuss doping-related issues with athletes. Athlete’s role: Passive. Name of the intervention (CG): Standard anti-doping education. Domain: Cognitive. Characteristics: A 60 min single-session using WADA’s NADO standard anti-doping education program. Athlete’s role: Passive.	Athletes Willingness to take prohibited substances (M ± SD) ↓* IG (T1 = 1.56 ± 0.99; T2 = 1.49 ± 0.85) vs. CG (T1 = 1.66 ± 1.10; T2 = 1.61 ± 1.01). ↔ IG (T2 = 1.49 ± 0.85; T3 = 1.42 ± 0.71) vs. CG (T2 = 1.61 ± 1.01; T3 = 1.57 ± 0.93). Doping moral disengagement (M ± SD) ↔ IG (T1 = 1.56 ± 0.77; T2 = 1.53 ± 0.83) vs. CG (T1 = 1.66 ± 0.85; T2 = 1.64 ± 0.87). ↔ IG (T2 = 1.53 ± 0.83; T3 = 1.53 ± 0.69) vs. CG (T2 = 1.64 ± 0.87; T3 = 1.57 ± 0.81). Attitudes toward doping (M ± SD) ↔ IG (T1 = 1.49 ± 0.74; T2 = 1.46 ± 0.75) vs. CG (T1 = 1.62 ± 0.82; T2 = 1.57 ± 0.91). ↔ IG (T2 = 1.46 ± 0.75; T3 = 1.46 ± 0.73) vs. CG (T2 = 1.57 ± 0.91; T3 = 1.55 ± 0.90). Efficacy to resist doping-related temptations (M ± SD) ↔ IG (T1 = 5.82 ± 1.83; T2 = 5.74 ± 1.94) vs. CG (T1 = 5.29 ± 2.08; T2 = 5.46 ± 2.05). ↔ IG (T2 = 5.74 ± 1.94; T3 = 5.79 ± 1.91) vs. CG (T2 = 5.46 ± 2.05; T3 = 5.41 ± 2.11). Behaviors against inadvertent doping (M ± SD) ↔ IG (T1 = 0.52 ± 1.16; T2 = 0.64 ± 1.16) vs. CG (T1 = 0.47 ± 0.99; T2 = 0.69 ± 1.28). ↔ IG (T2 = 0.64 ± 1.16; T3 = 0.58 ± 1.10) vs. CG (T2 = 0.69 ± 1.28; T3 = 0.74 ± 1.30). Anti-doping knowledge (M ± SD) ↔ IG (T1 = 2.53 ± 1.56; T2 = 2.73 ± 1.53) vs. CG (T1 = 2.51 ± 1.42; T2 = 2.56 ± 1.49). ↑* IG (T2 = 2.73 ± 1.53; T3 = 3.03 ± 1.49) vs. CG (T2 = 2.56 ± 1.49; T3 = 2.59 ± 1.58). Perceived coach need-related support (M ± SD) ↔ IG (T1 = 5.83 ± 0.78; T2 = 5.79 ± 0.88) vs. CG (T1 = 5.72 ± 0.82; T2 = 5.68 ± 0.84). ↔ IG (T2 = 5.79 ± 0.88; T3 = 5.39 ± 0.82) vs. CG (T2 = 5.68 ± 0.84; T3 = 5.87 ± 0.83). Perceived coach need-related thwarting (M ± SD) ↔ IG (T1 = 2.34 ± 0.88; T2 = 2.22 ± 0.92) vs. CG (T1 = 2.40 ± 0.93; T2 = 2.35 ± 0.90). ↔ IG (T2 = 2.22 ± 0.92; T3 = 2.08 ± 0.88) vs. CG (T2 = 2.22 ± 0.92; T3 = 2.16 ± 0.92). Basic psychological needs-related satisfaction (M ± SD) ↔ IG (T1 = 5.59 ± 0.98; T2 = 5.64 ± 0.94) vs. CG (T1 = 5.54 ± 0.85; T2 = 5.48 ± 0.90). ↔ IG (T2 = 5.64 ± 0.94; T3 = 5.77 ± 0.91) vs. CG (T2 = 5.48 ± 0.90; T3 = 5.62 ± 0.97). Basic psychological needs-related frustration (M ± SD) ↓* IG (T1 = 2.41 ± 1.19; T2 = 2.13 ± 0.97) vs. CG (T1 = 2.51 ± 1.20; T2 = 2.39 ± 1.13). ↔ IG (T2 = 2.13 ± 0.97; T3 = 1.13 ± 1.04) vs. CG (T2 = 2.39 ± 1.13; T3 = 2.32 ± 1.30).
Manges et al. [44], Germany and Austria	IG values (IGv) k = 14; *n* = 134 (41.8% ♀, 57.5% ♂). Age (M ± SD): 15.59 ± 1.54 y. IG information (IGi) k = 9; *n* = 114 (33.3% ♀, 65.8% ♂). Age (M ± SD): 15.38 ± 1.67 y. CG k = 7; *n* = 73 (64.4% ♀, 35.6% ♂). Age (M ± SD): 15.15 ± 1.60 y. Sport level: IGv: Regional = 23.9%; National = 50.7%; International = 23.9%; Other = 1.5%. IGi: Regional = 22.8%; National = 65.8%; International = 10.5%; Other = 0.9%. CG: Regional = 53.4%; National = 42.5%; International = 2.7%; Other = 1.4%.	Study design: Cluster randomized control trial with two conditions and control group: IGv, IGi, and CG. Three time measures: pre-intervention (IGv, IGi and CG), post-intervention (IGV, IGi and CG), and follow-up 4 months post-intervention (IGv and IGi). Measures: 1. Doping susceptibility; 2. Doping intention; 3. Moral disengagement; 4. Anticipated guilt; 5. Empathy; 6. Collective moral norms.	Name of the intervention (IGv): Values-based intervention. Domain: Cognitive and affective. Duration: Six 45 min sessions (one per week). Characteristics: 1. Doping, yes or no? (introduction); 2. Anticipated guilt; 3. Empathy; 4. Moral disengagement; 5. Collective moral norms; 6. Summary of all the topics and conclusion. Athlete’s role: Active. Name of the intervention (IGi): Information-based intervention. Domain: Cognitive. Duration: Six 45 min sessions (one per week). Characteristics: 1. Doping, what is it? (introduction); 2. The prohibited substances list; 3. Consequences of doping; 4. Doping control procedure; 5. Supplements and related risks; 6. Summary and internet resources. Athlete’s role: Active. CG: No intervention.	Doping susceptibility (M ± SD) ↔ IGv (T1 = 3.46 ± 2.06; T2 = 3.36 ± 1.95) vs. CG (T1 = 2.91 ± 1.84; T2 = 3.56 ± 1.67). ↔ IGi (T1 = 3.39 ± 2.08; T2 = 3.33 ± 2.17) vs. CG (T1 = 2.91 ± 1.84; T2 = 3.56 ± 1.67). IGv ↔ T1 (3.46 ± 2.06) vs. T2 (3.36 ± 1.95). IGv ↔ T2 (3.36 ± 1.95) vs. T3 (3.48 ± 2.08). IGi ↔ T1 (3.39 ± 2.08) vs. T2 (3.33 ± 2.17). IGi ↔ T2 (3.33 ± 2.17) vs. T3 (3.42 ± 2.21). Doping intention (M ± SD) ↔ IGv (T1 = 2.17 ± 1.51; T2 = 2.05 ± 1.41) vs. CG (T1 = 2.11 ± 1.34; T2 = 2.45 ± 1.55). ↔ IGi (T1 = 1.72 ± 1.11; T2 = 1.72 ± 0.98) vs. CG (T1 = 2.11 ± 1.34; T2 = 2.45 ± 1.55). IGv ↔ T1 (2.17 ± 1.51) vs. T2 (2.05 ± 1.41). IGv ↔ T2 (2.05 ± 1.41) vs. T3 (2.05 ± 1.29). IGi ↔ T1 (1.72 ± 1.11) vs. T2 (1.72 ± 0.98). IGi ↔ T2 (1.72 ± 0.98) vs. T3 (1.56 ± 0.90). Moral disengagement (M ± SD) ↓* IGv (T1 = 2.05 ± 0.90; T2 = 1.71 ± 0.69) vs. CG (T1 = 1.91 ± 0.92; T2 = 1.95 ± 0.86). IGi (T1 = 1.91 ± 0.78; T2 = 1.66 ± 0.68) vs. CG (T1 = 1.91 ± 0.92; T2 = 1.95 ± 0.86). IGv ↓* T1 (2.05 ± 0.90) vs. T2 (1.71 ± 0.69). IGv ↔ T2 (1.71 ± 0.69) vs. T3 (2.05 ± 1.29). IGi ↓* T1 (1.91 ± 0.78) vs. T2 (1.66 ± 0.68). IGi ↔ T2 (1.66 ± 0.68) vs. T3 (1.56 ± 0.90). Anticipated guilt (M ± SD) ↑* IGv (T1 = 5.79 ± 1.40; T2 = 6.16 ± 0.99) vs. CG (T1 = 6.05 ± 1.20; T2 = 5.93 ± 1.46). ↔ IGi (T1 = 6.06 ± 1.23; T2 = 6.13 ± 1.33) vs. CG (T1 = 6.05 ± 1.20; T2 = 5.93 ± 1.46). IGv ↑* T1 (5.79 ± 1.40) vs. T2 (6.16 ± 0.99). IGv ↓* T2 (6.16 ± 0.99) vs. T3 (5.34 ± 0.87). IGi ↔ T1 (6.06 ± 1.23) vs. T2 (6.13 ± 1.33). IGi ↔ T2 (6.13 ± 1.33) vs. T3 (6.02 ± 1.82). Empathy (M ± SD) ↔ IGv (T1 = 3.49 ± 0.48; T2 = 3.45 ± 0.57) vs. CG (T1 = 3.48 ± 0.53; T2 = 3.65 ± 0.55). ↔ IGi (T1 = 3.42 ± 0.61; T2 = 3.47 ± 0.56) vs. CG (T1 = 3.48 ± 0.53; T2 = 3.65 ± 0.55). IGv ↔ T1 (3.49 ± 0.48) vs. T2 (3.45 ± 0.57). IGv ↑* T2 (3.45 ± 0.57) vs. T3 (3.82 ± 0.54). IGi ↔ T1 (=3.42 ± 0.61) vs. T2 (3.47 ± 0.56). IGi ↔ T2 (3.47 ± 0.56) vs. T3 (3.43 ± 0.73). Collective moral norms (M ± SD) ↔ IGv (T1 = 4.10 ± 0.70; T2 = 4.10 ± 0.88) vs. CG (T1 = 3.98 ± 0.86; T2 = 4.05 ± 0.74). ↔ IGi (T1 = 4.20 ± 0.73; T2 = 4.28 ± 0.69) vs. CG (T1 = 3.98 ± 0.86; T2 = 4.05 ± 0.74). IGv ↔ T1 (4.10 ± 0.70) vs. T2 (4.10 ± 0.88). IGv ↔ T2 (4.10 ± 0.88) vs. T3 (4.19 ± 0.79). IGi ↔ T1 (=4.20 ± 0.73) vs. T2 (4.28 ± 0.69). IGi ↔ T2 (4.28 ± 0.69) vs. T3 (4.24 ± 0.67).
Nicholls et al. [45], United Kingdom (UK)	IG Presentation (IGp) *n* = 254 (6.3% ♀, 93.7% ♂). Age (M ± SD): 16.5 ± 1.1 y. IG Online (IGo) *n* = 251 (20.7% ♀, 79.3% ♂). Age (M ± SD): 15.9 ± 1.3 y. IG Presentation Online (IGo-p) *n* = 262 (100% ♂). Age (M ± SD): 16.2 ± 1.3 y. CG *n* = 314 (34.7% ♀, 65.3% ♀). Age (M ± SD): 15.9 ± 1.6 y. Sport level: High-level adolescent athletes.	Study design: Cluster randomized controlled trial with three conditions and control group: IGp, IGo, IGo-p, and CG. Three time measures: pre, post-intervention, and follow-up (8 weeks after intervention). Measures: The Adolescent Sport Doping Inventory two dimensions: 1. Doping attitudes; 2. Doping susceptibility.	Name of intervention (IGp): Presentation iPlayClean. Domain: Cognitive and affective. Duration: Two 90 min face-to-face sessions for athletes (8 weeks between sessions) and one 60 min face-to-face session for parents and coaches for IG presentation. Characteristics: (1) Introduction to doping; (2) Goals; (3) Motivation; (4) Doping myths. (5) Playing fair; (6) Resisting temptations; (7) Making the right decisions; (8) Drug testing and health; (9) Nutritional supplements; (10) Coping strategies. Athlete’s role: Active. Name of intervention (IGo): Online iPlayClean. Domain: Cognitive and affective. Duration: Online access to IPlayClean website. Characteristics: Free interaction on the IPlayClean web platform. Athlete’s role: Active. Name of intervention (IGp-o): Online presentation iPlayClean. Domain: Cognitive and affective. Duration: Online access to IPlayClean website and two 90 min face-to-face sessions for athletes (8 weeks between sessions). Sixty minute face-to-face session for parents and coaches for IG presentation. Characteristics: Free interaction on the IPlayClean web platform and 10 of the same modules as the presentation. Athlete’s role: Active. CG: No intervention	Doping attitudes (M ± SD) ↓* IGp (T1 = 10.50 ± 7.30; T2 = 5.80 ± 2.70) vs. CG (T1 = 9.80 ± 5.90; T2 = 10.70 ± 6.70). ↓* IGo (T1 = 11.10 ± 6.70; T2 = 6.00 ± 3.20) vs. CG (T1 = 9.80 ± 5.90; T2 = 10.70 ± 6.70). ↓* IGo-p (T1 = 9.30 ± 6.30; T2 = 6.40 ± 3.10) vs. CG (T1 = 9.80 ± 5.90; T2 = 10.70 ± 6.70). ↓* IGp (T1 = 10.50 ± 7.30; T3 = 4.90 ± 1.50) vs. CG (T1 = 9.80 ± 5.90; T3 = 9.90 ± 6.20). ↓* IGo (T1 = 11.10 ± 6.70; T3 = 6.40 ± 3.10) vs. CG (T1 = 9.80 ± 5.90; T3 = 9.90 ± 6.20). ↓* IGo-p (T1 = 9.30 ± 6.30; T3 = 6.60 ± 3.00) vs. CG (T1 = 9.80 ± 5.90; T3 = 9.90 ± 6.20). Doping susceptibility (M ± SD) ↓* IGp (T1 = 9.70 ± 6.60; T2 = 7.50 ± 3.70) vs. CG (T1 = 12.50 ± 8.20; T2 = 12.20 ± 7.30). ↓* IGo (T1 = 12.30 ± 8.10; T2 = 9.10 ± 5.40) vs. CG (T1 = 12.50 ± 8.20; T2 = 12.20 ± 7.30). ↓* IGo-p (T1 = 15.20 ± 10.40; T2 = 8.70 ± 4.40) vs. CG (T1 = 12.50 ± 8.20; T2 = 12.20 ± 7.30). ↓* IGp (T1 = 9.70 ± 6.60; T3 = 6.30 ± 2.70) vs. CG (T1 = 12.50 ± 8.20; T3 = 12.50 ± 7.90). ↔ IGo (T1 = 12.30 ± 8.10; T3 = 9.80 ± 6.10) vs. CG (T1 = 12.50 ± 8.20; T3 = 12.50 ± 7.90). ↔ IGo-p (T1 = 15.20 ± 10.40; T3 = 10.90 ± 6.30) vs. CG (T1 = 12.50 ± 8.20; T3 = 12.50 ± 7.90).
Kavussanu et al. [23], United Kingdom (UK) and Greece	*United Kingdom (UK)* IG Moral (IGm) k = 6; *n* = 66 (24.2% ♀, 75.8% ♂). Age (M ± SD): 16.61 ± 0.68 y. IG Education (IGe) k = 6; *n* = 55 (38.2% ♀, 61.8% ♂). Age (M ± SD): 18.00 ± 1.83 y. *Greece (Gr)* IGm k = 10; *n* = 102 (35.3% ♀, 64.7% ♂). Age (M ± SD): 18.19 ± 2.49 y. IGe k = 11; *n* = 80 (36.3% ♀, 63.7% ♂). Age (M ± SD): 19.16 ± 1.69 y Sport level: Not specified.	Study design: Cluster randomized control trial with two conditions (moral and intervention) and two countries (UK and Greece). Four time point measures: pre-test, post-test, 3 months post-test, and 6 months post-test. Measures: 1. Doping likelihood; 2. Moral identity; 3. Moral disengagement; 4. Moral atmosphere; 5. Anticipated guilt.	Name of intervention: Moral intervention. Domain: Affective. Duration: Six 1 h sessions. Characteristics: (a) Moral identity: 1. Success in sport. 2. Values in sport. (b) Moral disengagement: 3. Justification for doping. 4. Consequences of doping for others. (c) Moral atmosphere: 5. The culture of the team. 6. Course conclusion. Athlete’s role: Active. Name of intervention: Educational intervention. Domain: Cognitive. Duration: Six 1 h sessions. Characteristics: 1. Introduction to doping; 2. Doping control; 3. Banned substances; 4. Sport supplements; 5. Nutrition; 6. Whistleblowing. Athlete’s role: Active.	Doping likelihood (M ± SD) IGm ↓* T1 (UK: 2.37 ± 1.49; Gr: 2.36 ± 1.40) vs. T2 (UK: 1.97 ± 1.22; Gr: 1.73 ± 0.97). IGm ↔ T2 (UK: 1.97 ± 1.22; Gr: 1.73 ± 0.97) vs. T3 (UK: 1.83 ± 1.13; Gr: 1.68 ± 0.92). IGm ↔ T3 (UK: 1.83 ± 1.13; Gr: 1.68 ± 0.92) vs. T4 (UK: 1.68 ± 1.13; Gr: 1.54 ± 0.82). IGe ↓* T1 (UK: 2.88 ± 1.69; Gr: 2.30 ± 1.17) vs. T2 (UK: 2.37 ± 1.56; Gr: 1.94 ± 1.13). IGe ↔ T2 (UK: 2.37 ± 1.56; Gr: 1.94 ± 1.13) vs. T3 (UK: 2.00 ± 1.39; Gr: 1.96 ± 1.21). IGe ↔ T3 (UK: 2.00 ± 1.39; Gr: 1.96 ± 1.21) vs. T4 (UK: 1.85 ± 1.12; Gr: 2.13 ± 1.45). Moral identity (M ± SD) IGm ↔ T1 (UK: 5.65 ± 1.16; Gr: 5.76 ± 1.19) vs. T2 (UK: 5.62 ± 1.21; Gr: 6.23 ± 0.96). IGm ↔ T2 (UK: 5.62 ± 1.21; Gr: 6.23 ± 0.96) vs. T3 (UK: 5.66 ± 1.12; Gr: 6.18 ± 0.82). IGm ↔ T3 (UK: 5.66 ± 1.12; Gr: 6.18 ± 0.82) vs. T4 (UK: 5.81 ± 1.26; Gr: 6.27 ± 0.70). IGe ↔ T1 (UK: 5.86 ± 1.08; Gr: 5.91 ± 0.95) vs. T2 (UK: 6.24 ± 0.84; Gr: 6.00 ± 0.96). IGe ↔ T2 (UK: 6.24 ± 0.84; Gr: 6.00 ± 0.96) vs. T3 (UK: 6.15 ± 1.05; Gr: 6.00 ± 0.84). IGe ↔ T3 (UK: 6.15 ± 1.05; Gr: 6.00 ± 0.84) vs. T4 (UK: 6.12 ± 1.01; Gr: 5.72 ± 1.12). Moral disengagement (M ± SD) IGm ↓* T1 (UK: 2.37 ± 0.87; Gr: 1.91 ± 0.78) vs. T2 (UK: 2.15 ± 0.88; Gr: 1.71 ± 0.75). IGm ↔ T2 (UK: 2.15 ± 0.88; Gr: 1.71 ± 0.75) vs. T3 (UK: 2.09 ± 1.10; Gr: 1.62 ± 0.64). IGm ↔ T3 (UK: 2.09 ± 1.10; Gr: 1.62 ± 0.64) vs. T4 (UK: 1.92 ± 0.93; Gr: 1.49 ± 0.60). IGe ↓* T1 (UK: 2.47 ± 0.88; Gr: 2.23 ± 1.05) vs. T2 (UK: 2.07 ± 0.66; Gr: 1.73 ± 0.68). IGe ↔ T2 (UK: 2.07 ± 0.66; Gr: 1.73 ± 0.68) vs. T3 (UK: 2.09 ± 0.89; Gr: 1.85 ± 0.83). IGe ↔ T3 (UK: 2.09 ± 0.89; Gr: 1.85 ± 0.83) vs. T4 (UK: 1.86 ± 0.74; Gr: 1.81 ± 0.79). Moral atmosphere (M ± SD) IGm ↔ T1 (UK: 2.34 ± 1.01; Gr: 2.61 ± 1.11) vs. T2 (UK: 2.30 ± 1.08; Gr: 2.25 ± 0.89). IGm ↔ T2 (UK: 2.30 ± 1.08; Gr: 2.25 ± 0.89) vs. T3 (UK: 1.94 ± 0.92; Gr: 2.11 ± 0.88). IGm ↔ T3 (UK: 1.94 ± 0.92; Gr: 2.11 ± 0.88) vs. T4 (UK: 1.96 ± 1.00; Gr: 2.02 ± 0.81). IGe ↔ T1 (UK: 2.78 ± 1.37; Gr: 2.77 ± 1.06) vs. T2 (UK: 2.45 ± 1.09; Gr: 2.53 ± 0.99). IGe ↔ T2 (UK: 2.45 ± 1.09; Gr: 2.53 ± 0.99) vs. T3 (UK: 2.06 ± 0.98; Gr: 2.65 ± 1.19). IGe ↔ T3 (UK: 2.06 ± 0.98; Gr: 2.65 ± 1.19) vs. T4 (UK: 1.90 ± 0.87; Gr: 2.37 ± 1.00). Anticipated guilt (M ± SD) IGm ↑* T1 (UK: 5.34 ± 1.45; Gr: 5.37 ± 1.28) vs. T2 (UK: 5.56 ± 1.48; Gr: 5.68 ± 1.28). IGm ↔ T2 (UK: 5.56 ± 1.48; Gr5.68 ± 1.28) vs. T3 (UK: 5.84 ± 1.42; Gr: 5.74 ± 1.24) IGm ↔ T3 (UK: 5.84 ± 1.42; Gr: 5.74 ± 1.24) vs. T4 (UK: 5.83 ± 1.62; Gr: 5.54 ± 1.46). IGe ↑* T1 (UK: 5.24 ± 1.51; Gr: 5.07 ± 1.45) vs. T2 (UK: 5.80 ± 1.19; Gr: 5.32 ± 1.41). IGe ↔ T2 (UK: 5.80 ± 1.19; Gr: 5.32 ± 1.41) vs. T3 (UK: 5.94 ± 1.41; Gr: 5.32 ± 1.40). IGe ↔ T3 (UK: 5.94 ± 1.41; Gr: 5.32 ± 1.40) vs. T4 (UK: 5.99 ± 1.34; Gr: 5.41 ± 1.43).
Kavussanu et al. [16], United Kingdom (UK) and Greece	IG Psychological (IGp) k = 10; *n* = 109 (38.5% ♀, 58.7% ♂; 2.8% not specified). Age (M ± SD): 17.68 ± 1.76 y. IG Education (IGe) k = 9; *n* = 99 (48.5% ♀, 51.5% ♂). Age (M ± SD): 18.54 ± 2.51 y. Sport level: Not specified.	Study design: Cluster randomized control trial with two conditions (psychological and intervention) and parallel group (UK and Greece). Three time point measures: pre-test, post-test, and 2 month follow-up. Measures: 1. Doping likelihood; 2. Anticipated guilt; 3. Moral disengagement; 4. Self-regulatory efficacy.	Name of intervention: Psychological intervention. Domain: Affective. Duration: Six 1 h sessions. Characteristics: 1. Moral agency; 2. Emotions; 3. Moral disengagement; 4. Moral engagement; 5. Self-regulatory efficacy; 6. Course review and conclusion. Athlete’s role: Active. Name of intervention: Educational intervention. Domain: Cognitive. Duration: Six 1 h sessions. Characteristics: 1. Introduction to WADA and regulation of doping in sport; 2. Doping control process; 3. Prohibited substances and their side effects; 4. The risks of supplements; 5. The role of healthy nutrition in terms of benefiting performance and recovery; 6. Whistleblowing and its role in protecting clean athletes. Athlete’s role: Active.	Doping likelihood (M ± SD) IGp ↓* T1 (2.53 ± 1.62) vs. T2 (1.76 ± 1.05). IGp ↔ T2 (1.76 ± 1.05) vs. T3 (1.76 ± 0.96). IGe ↔ T1 (2.51 ± 1.51) vs. T2 (2.15 ± 1.36). IGe ↓* T2 (2.15 ± 1.36) vs. T3 (1.88 ± 1.24). T1 to T2 changes ↓* IGp (2.53 ± 1.62; 1.76 ± 1.05) vs. IGe (2.51 ± 1.51; 2.15 ± 1.36). T2 to T3 changes ↓* IGe (2.15 ± 1.36; 1.88 ± 1.24) vs. IGp (1.76 ± 1.05; 1.76 ± 0.96). Anticipated guilt (M ± SD) IGp (pp) ↑* T1 (4.85 ± 2.11) vs. T2 (5.82 ± 1.54). IGp ↑* T2 (5.82 ± 1.54) vs. T3 (6.41 ± 0.83). IGe ↔ T1 (4.95 ± 1.74) vs. T2 (5.36 ± 1.56). IGe ↑* T2 (5.36 ± 1.56) vs. T3 (5.45 ± 1.62). T1 to T2 changes ↑* IGp (4.85 ± 2.11; 5.82 ± 1.54) vs. IGe (4.95 ± 1.74; 5.36 ± 1.56). T2 to T3 changes ↓* IGp (5.82 ± 1.54; 6.41 ± 0.83) vs. IGe (5.36 ± 1.56; 5.45 ± 1.62). Moral disengagement (M ± SD) IGp ↓* T1 (2.17 ± 0.99) vs. T2 (1.83 ± 0.74). IGp ↓* T2 (1.83 ± 0.74) vs. T3 (1.66 ± 0.72). IGe ↓* T1 (2.34 ± 1.07) vs. T2 (1.95 ± 0.98). IGe ↔ T2 (1.95 ± 0.98) vs. T3 (1.96 ± 0.92). T1 to T2 changes ↓* IGp (2.17 ± 0.99; 1.83 ± 0.74) vs. IGe (2.34 ± 1.07; 1.95 ± 0.98). T2 to T3 changes ↔ IGp (1.83 ± 0.74; 1.66 ± 0.72) vs. IGe (1.95 ± 0.98; 1.96 ± 0.92). Self-regulatory efficacy (mean ± SD) IGp (pp) ↓* T1 (5.39 ± 1.55) vs. T2 (5.06 ± 1.95). IGp ↔ T2 (5.06 ± 1.95) vs. T3 (5.43 ± 1.87). IGe ↔ T1 (4.76 ± 1.84) vs. T2 (5.17 ± 1.80). IGe ↔ T2 (5.17 ± 1.80) vs. T3 (5.29 ± 1.65). T1 to T2 changes ↑* IGe (4.76 ± 1.84; 5.17 ± 1.80) vs. IGp (5.39 ± 1.55; 5.06 ± 1.95). T2 to T3 changes ↔ IGe (5.17 ± 1.80; 5.29 ± 1.65) vs. IGp (5.06 ± 1.95; 5.43 ± 1.87).
Galli et al. [46], Italy	IG *n* = 167 (37.7% ♀, 62.3% ♂) Age (M ± SD): 17.51 ± 0.58 y. CG *n* = 112 (42% ♀, 58% ♂). Age (M ± SD): 17.65 ± 1.0 y. Sport level: IG: Amateur = 10.2%; Local = 3.6%; Regional = 37.1%; National = 38.9%; International = 6.0%; Not specified = 4.2%. *CG*: Amateur = 15.2%; Local = 13.4%; Regional = 35.7%; National = 19.6%; International = 7.1%; Not specified = 8.9%.	Study design: Quasi-experimental longitudinal design (two groups). Pre-test and post-test measures for both groups (intervention and control). Measures: 1. Doping intention; 2. Self-regulatory efficacy to resist social pressure towards the use of substances; 3. Moral disengagement; 4. Doping knowledge.	Name of the intervention: “Serious Game” (virtual video game). Domain: Cognitive and affective. Duration: Four 90 min sessions (one per month). Characteristics: Simulation of track and field athlete’s everyday life, four weeks before relevant competition. 1. Introduction to the “serious game”; 2. Play and discussion about the “serious game”; 3. Discussion about results of the serious game; 4. Doping knowledge class and discussion. Athlete’s role: Active. CG: No intervention.	Doping intention (M ± SD) ↔ IG T1 (=2.72 ± 1.39) vs. T2 (2.57 ± 1.44). ↔ CG T1 (2.51 ± 1.37) vs. T2 (2.67 ± 1.47). ↓* IG (T1 = 2.72 ± 1.39; T2 = 2.57 ± 1.44) vs. CG (T1 = 2.51 ± 1.37; T2 = 2.67 ± 1.47). Self-regulatory efficacy to resist social pressure towards the use of substances (M ± SD) ↓* IG T1 (5.52 ± 1.54) vs. T2 (5.12 ± 1.86). ↓* CG T1 (5.28 ± 1.82) vs. T2 (4.89 ± 1.88). ↔ IG (T1 = 5.52 ± 1.54; T2 = 5.12 ± 1.86) vs. CG (T1 = 5.28 ± 1.82; T2 = 4.89 ± 1.88). Moral disengagement (M ± SD) ↓* IG T1 (1.77 ± 0.51) vs. T2 (1.61 ± 0.52). ↓* CG T1 (1.64 ± 0.64) vs. T2 (1.57 ± 0.62). ↔ IG (T1 = 1.77 ± 0.51; T2 = 1.61 ± 0.52) vs. CG (T1 = 1.64 ± 0.64; T2 = 1.57 ± 0.62). Doping knowledge (M ± SD) ↔ IG T1 (6.59 ± 1.37) vs. T2 (7.16 ± 1.38). ↔ CG T1 (6.58 ± 1.74) vs. T2 (6.42 ± 1.92). ↑* IG (T1 = 6.59 ± 1.37; T2 = 7.16 ± 1.38) vs. CG (T1 = 6.58 ± 1.74; T2 = 6.42 ± 1.92). “Serious game” mediation effect Doping intention T1 → serious game scores → doping intention T2 (*β* = 0.13, 95% CI [0.06, 0.23]).
Deng et al. [48], China	IG athletes *n* = 16 (50% ♀, 50% ♂). Age (M ± SD): 20.7 ± 2.33 y. IG non-athletes *n* = 16 (50% ♀, 50% ♂). Age (M ± SD): 22.5 ± 2.94 y. CG athletes *n* = 16 (50% ♀, 50% ♂). Age (M ± SD): 21.81 ± 3.12 y. CG non-athletes *n* = 16 (50% ♀, 50% ♂). Age (M ± SD): 24.75 ± 3.64 y. Sport level: Intervention group: College first-level athletes = 50%; Non-athletes = 50%. Control group: College first-level athletes = 50%; Non-athletes = 50%.	Study design: Quasi-experimental design (intervention and control group). Pre-test–post-test measures. Measures: 1. ALPHA test (doping knowledge); 2. Performance enhancement attitude scale (attitudes towards doping); 3. Doping likelihood; 4. Picture-based doping brief implicit association test (BIAT, reaction time and error rate for judging pictures about doping and healthy food); 5. Functional near-infrared spectroscopy to monitor BIAT tasks.	Name of the intervention: Athlete Learning Program about Health and Anti-Doping (ALPHA). Domain: Cognitive. Duration: A single session to learn the eight ALPHA modules. All interventions had a mean 80 min duration. Characteristics: Eight modules: 1. The doping control process; 2. Whereabouts of athletes; 3. Therapeutic use exemptions (TUEs); 4. Results management; 5. Medical reasons not to dope; 6. Ethical reasons not to dope; 7. Practical help to stay clean; 8. How to deal with pressure. Athlete’s role: Passive. CG: No intervention.	Doping knowledge (M ± SD) IG ↑* T1 (non-athlete = 9.50 ± 1.46; athlete = 10.13 ± 1.31) vs. T2 (non-athlete = 11.19 ± 0.75; athlete = 11.13 ± 1.02). CG ↔ T1 (non-athlete = 9.06 ± 2.18; athlete = 10.06 ± 1.12) vs. T2 (non-athlete = 8.94 ± 2.02; athlete = 10.12 ± 1.26). IG ↔ athletes (T1 = 10.13 ± 1.31; T2 = 11.13 ± 1.02) vs. non-athletes (T1 = 9.50 ± 1.46; T2 = 11.19 ± 0.75). CG ↔ athletes (T1 = 10.06 ± 1.12; T2 = 10.12 ± 1.26) vs. non-athletes (T1 = 9.06 ± 2.18; T2 = 8.94 ± 2.02). Attitudes towards doping (M ± SD) IG ↓* T1 (non-athlete = 30.50 ± 9.68; athlete = 26.75 ± 7.50) vs. T2 (non-athlete = 25.56 ± 12.48; athlete = 23.00 ± 10.28). CG ↔ T1 (non-athlete = 29.43 ± 11.09; athlete = 27.37 ± 8.48) vs. T2 (non-athlete = 28.50 ± 11.46; athlete = 27.75 ± 8.72). IG ↔ athletes (T1 = 26.75 ± 7.50; T2 = 26.75 ± 7.50) vs. non-athletes (T1 = 30.50 ± 9.68; T2 = 25.56 ± 12.48). CG ↔ athletes (T1 = 27.37 ± 8.48; T2 = 27.75 ± 8.72) vs. non-athletes (T1 = 29.43 ± 11.09; T2 = 28.50 ± 11.46). Doping likelihood benefit situation (M ± SD) IG ↔ T1 (non-athlete = 2.74 ± 1.38; athlete = 1.97 ± 0.98) vs. T2 (non-athlete = 2.28 ± 1.19; athlete = 1.61 ± 0.80). CG ↔ T1 (non-athlete = 2.72 ± 1.22; athlete = 2.03 ± 0.92) vs. T2 (non-athlete = 2.68 ± 1.30; athlete = 2.10 ± 1.28). IG ↓* athletes (T1 = 1.97 ± 0.98; T2 = 1.61 ± 0.80) vs. non-athletes (T1 = 2.74 ± 1.38; T2 = 2.28 ± 1.19). CG ↓* athletes (T1 = 2.03 ± 0.92; T2 = 2.10 ± 1.28) vs. non-athletes (T1 = 2.72 ± 1.22; T2 = 2.68 ± 1.30). Doping likelihood cost situation (M ± SD) IG ↔ T1 (non-athlete = 1.51 ± 1.02; athlete = 1.25 ± 0.52) vs. T2 (non-athlete = 1.34 ± 0.53; athlete = 1.20 ± 0.50). CG ↔ T1 (non-athlete = 1.33 ± 0.43; athlete = 1.31 ± 0.50) vs. T2 (non-athlete = 1.28 ± 0.49; athlete = 1.41 ± 0.65). IG ↔ athletes (T1 = 1.25 ± 0.52; T2 = 1.20 ± 0.50) vs. non-athletes (T1 = 1.51 ± 1.02; T2 = 1.34 ± 0.53). CG ↔ athletes (T1 = 1.31 ± 0.50; T2 = 1.41 ± 0.65) vs. non-athletes (T1 = 1.33 ± 0.43; T2 = 1.28 ± 0.49).
Hurst et al. [47], United Kingdom (UK)	IG *n* = 302 (41.4% ♀, 58.6% ♂). Age (M ± SD): 18.71 ± 2.61 y. Sport level: *IG*: Regional = 37.8%; National = 32.7%; International = 29.5%.	Study design: Pragmatic within-participant pre/post design (pre-experimental longitudinal design). Pre-test and post-test measures (three months after intervention). Measures: 1. Doping susceptibility; 2. Intention to use dietary supplements; 3. Spirit of sport; 4. Moral values; 5. Knowledge on anti-doping; 6. Anti-doping practices; 7. Whistleblowing.	Name of the intervention: UK Anti-Doping Clean Sport education program. Domain: Cognitive. Duration: A 60 min single session. Characteristics: Participants were provided with an overview of the doping control testing procedures and their rights and responsibilities under the World Anti-Doping Code. Athlete’s role: Passive.	Doping susceptibility (M ± SD) ↓* IG T1 (1.58 ± 1.35) vs. T2 (1.48 ± 1.20). Dietary supplementation intention (M ± SD) ↓* IG T1 (4.10 ± 2.32) vs. T2 (3.55 ± 2.25) Spirit of sport values (M ± SD) ↑* IG T1 (4.17 ± 0.85) vs. T2 (4.42 ± 0.80). Moral values (M ± SD) ↑* IG T1 (4.29 ± 0.50) vs. T2 (4.52 ± 0.44). Anti-doping knowledge (M ± SD) ↑* IG T1 (4.07 ± 1.26) vs. T2 (5.57 ± 1.19). Anti-doping practices (M ± SD) ↑* IG T1 (2.38 ± 1.04) vs. T2 (3.43 ± 1.25). Whistleblowing (M ± SD) ↑* IG T1 (3.99 ± 1.39) vs. T2 (5.81 ± 1.28). Anti-doping program mediation effect Doping susceptibility T1 → Δ supplement intention → doping susceptibility T2: (*β* = 0.04, 95% CI [0.01 to 0.07]). Doping susceptibility T1 → Δ supplement intention → supplement intention x moral values → doping susceptibility T2: (*β*_1_ = 0.05, 95% CI [0.01, 0.12]; *β*_2_ = 0.04, 95% CI [0.01, 0.09]; *β*_1_ = 0.03, 95% CI [−0.03, 0.10]).
da Silva et al. [40], Brazil	IG game: *n* = 20 (100% ♂) Age range: 18–20 y. Sport level: Semi-professional soccer players.	Study design: Quasi-experimental and descriptive design, one group with pre/post-test intervention measures. Measures: Ad hoc questionnaire design by the authors to measure: 1. Positive factors: non-steroidal anti-inflammatory, low dose of caffeine, vitamin C, doping, fruit, physical exercise, and water (20 items); 2. Negative factors: erythropoietin, diuretics, ephedrine (>10 mcg), contaminated thermogenic, growth hormone (GH), anabolic steroids, contaminated supplementation, cocaine, testosterone, and cannabis (20 items).	Name of intervention: Heart in Game. Domain: Cognitive. Duration: A single session to play the game. The game duration was 6 min as the maximum time and consisted of three stages. Characteristics: Athletes must play three stages of the game which presents to the athlete different scenarios where they must accomplish the mission to collect items considered positive factors and avoid the negative factors. The scenarios were: (1) a pharmacy; (2) a supermarket; (3) a gym. Athlete’s role: Active.	Positive factor knowledge (% of correct answers) Low dose of caffeine ↑* T1 (30%) vs. T2 (70%). Doping knowledge ↑* T1 (20%) vs. T2 (75%). Vitamin C ↔ T1 (95%) vs. T2 (90%). Physical activity ↔ T1 (95%) vs. T2 (100%). Non-steroidal anti-inflammatory ↔ T1 (40%) vs. T2 (40%). Fruits ↔ T1 (100%) vs. T2 (100%). Water ↔ T1 (100%) vs. T2 (100%). Negative factor knowledge (% of correct answers) Diuretics ↑* T1 (25%) vs. T2 (50%). Contaminated thermogenic ↑* T1 (35%) vs. T2 (70%). Growth hormone ↑* T1 (20%) vs. T2 (70%). Contaminated supplements ↑* T1 (15%) vs. T2 (80%). Testosterone ↑* T1 (35%) vs. T2 (90%). Erythropoietin ↔ T1 (20%) vs. T2 (45%). Ephedrine (>10 mcg) ↔ T1 (50%) vs. T2 (65%). Cannabis ↔ T1 (90%) vs. T2 (100%). Anabolic steroids ↔ T1 (90%) vs. T2 (80%). Cocaine ↔ T1 (85%) vs. T2 (85%).
Thomas et al. [39], United Arab Emirates (UAE)	IG *n* = 218 (100% ♂). Age (M ± SD): Not specified. Sport level: Adult non-sports employment (amateur) = 36.7%; full-time bodybuilders (professional) = 7.3%. University student bodybuilders = 50.9%. School student bodybuilders *n* = 5.0%.	Study design: Pre-experimental design with pre/post-test intervention measures. Measures: 1. Modified version (8 items) of Performance Enhancement Attitude Scale (PEAS).	Name of intervention: Educational Flyer “The Power of the Right Choice”. Domain: Cognitive. Duration: One session. Characteristics: An anti-doping educational flyer with a clear message about the risk of doping, information about how to check for safe supplements, and resources to learn about doping prevention. Athlete’s role: Passive.	Performance Enhancement Attitude Scale (median) ↓* T1 (median = 24) vs. T2 (median = 14). School student bodybuilders ↓* (median T2 = 11) vs. Full-time bodybuilders T2 (median = 21).

Note. ♀: women; ♂: men; y: years; IG: intervention group; CG: control group; M: mean; SD: standard deviation; k: number of groups; ↓: decrease; ↑: increase; ↔: without change; *: statistically significant change (*p* < 0.05); →: conditional sequence; *β*: Beta coefficient; CI: confidence interval; Δ: change; pp: per protocol analysis; T1: evaluation before intervention program (pre); T2: evaluation after intervention program (post); T3: follow-up number 1 after post-intervention evaluation; T4: follow-up number 2 after post-intervention evaluation.

**Table 4 sports-13-00108-t004:** Meta-regression coefficients (statistically significative).

		Pre–Post-Test			Pre-Test–Follow-Up		
Variable	Moderator	ES	95% CI	*p*	ES	95% CI	*p*
Doping Intention	Athlete’s role						
	Passive	−0.83	−1.128, −0.534	<0.001	−0.86	−1.228, −0.487	<0.001
	Design						
	Quasi-experimental	−0.76	−1.072, −0.441	<0.001			
	Age	0.12	0.061, 0.183	<0.001	0.13	0.058, 0.210	0.002
	Number of sessions						
	Long (six or more)	−0.55	−0.773, −0.320	<0.001	−0.48	−0.762, −0.204	0.002
Anti-Doping Behavior	Athlete’s role						
	Passive	−0.77	−1.374, −0.162	<0.019			

Note. ES: effect size (Cohen’s d); CI: confidence interval; *p*: *p*-value of significance.

## Data Availability

All the data generated or analyzed during this study are included in the article as Table(s), Figure(s), and/or Electronic Appendix A. Any other data requirement can be directed to the corresponding author upon reasonable request.

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
