# Peer review of "Effective Intervention Features of a Doping Prevention Program for Athletes: A Systematic Review with Meta-Analysis"

_sports, 2025, doi:10.3390/sports13040108_

Round 1
Reviewer 1 Report
Comments and Suggestions for Authors
Thank you for the opportunity to review this article. This is a very interesting article and a useful addition in the doping literature. Below there are some comments that the authors may find useful in improving the quality of the article.
A concern I have is the distinction between cognitive and affective approaches to doping prevention. Although I understand the rationale for this distinction, this is not conceptually correct. More specifically the theoretical models assumed to describe the concepts in the affective approach are actually cognitive, i.e., social cognitive theory by Bandura.
In this line, the variables included in this approach are rather cognitive and not affective. For instance both motivation and moral disengagement are cognitive variables and not affective.
In addition, attitudes are reported as both part of the cognitive and affective approach. In fact, attitudes may have a cognitive and affective component but these are not measured separately and a total score of attitudes has been used in the presented interventions. Overall, I think that a restructuring is needed in the introduction to avoid the cognitive-affective distinction.
In this respect reference to other reviews and meta-analyses would be useful to set the basis of the study (e.g., Ntoumanis et al., 2014, 2024, Barnes et al., 2019 etc).
In line with the above the study needs a stronger rationale. For example how this study extends our understanding or anti-doping interventions beyond the Ntoumanis et al. (2024) meta-analysis?
In the method section the authors need to clarify the selected timeframe. Why the studied the period 2019-2024 and not a larger timeframe?
In the discussion section, again I think that the cognitive-affective distinction should be avoided.
Also, I think that the discussion would benefit from a more thorough interpretation of the findings. As it is, it largely reiterates the results of the review. I think a more thorough discussion on the what worked well and why or what didn’t worked well and why would be really meaningful and would add to visibility of the article. In addition, which variables were affected more by the interventions, which ones of them have been found more crucial in defining doping behavior etc, are important questions that the review could provide meaningful insights to anti-doping research.
In line with the above, I think a section with practical implications derived from the review would be useful to the readers.
Author Response
Dear Journal Editor of Sports,
I would like to begin by expressing my gratitude for the opportunity to contribute to this manuscript for such a prestigious journal. I also want to extend my sincere appreciation to the three reviewers who evaluated the manuscript—their commitment and expertise have undoubtedly played a crucial role in enhancing its quality.
Below, I provide a detailed response to each of the reviewers' observations, comments, and suggestions. I outline the corresponding explanations, the revisions made, and the decisions taken to improve the manuscript based on their valuable feedback.
Reviewer 1.
A concern I have is the distinction between cognitive and affective approaches to doping prevention. Although I understand the rationale for this distinction, this is not conceptually correct. More specifically the theoretical models assumed to describe the concepts in the affective approach are actually cognitive, i.e., social cognitive theory by Bandura.
In this line, the variables included in this approach are rather cognitive and not affective. For instance both motivation and moral disengagement are cognitive variables and not affective.
In addition, attitudes are reported as both part of the cognitive and affective approach. In fact, attitudes may have a cognitive and affective component but these are not measured separately and a total score of attitudes has been used in the presented interventions. Overall, I think that a restructuring is needed in the introduction to avoid the cognitive-affective distinction.
Authors response.
Thank you for your observation. The distinction between cognitive and affective intervention approaches is based on the classification established by the International Standard for Education of WADA. This classification highlights the key components of each approach: the cognitive approach focuses on information, knowledge, and consequences related to health and punishment, while the affective approach emphasizes personal values that foster fair play, personal integrity, and respect for others and sport itself.
Revisions were made to refine the wording, particularly in lines 42-47 and 50-55, to better emphasize this distinction. Additionally, the theories supporting the cognitive approach in anti-doping interventions were explained in greater detail, with particular focus on social learning theories and the health belief model. This expanded explanation can be found in lines 77-86. Similarly, to address potential confusion surrounding the justification of the theory of planned behavior, adjustments were made in lines 105-108 to reinforce the affective approach in anti-doping education.
Reviewer 1.
In this respect reference to other reviews and meta-analyses would be useful to set the basis of the study (e.g., Ntoumanis et al., 2014, 2024, Barnes et al., 2019 etc).
In line with the above the study needs a stronger rationale. For example how this study extends our understanding or anti-doping interventions beyond the Ntoumanis et al. (2024) meta-analysis?
Authors response.
Thank you very much for your suggestion, we add a brief explanation about the influence of the conclusions by Ntoumanis et al. (2024) and pointed out the way that our manuscript could help to extend the understanding of the effectiveness on doping prevention intervention programs.
Reviewer 1.
In the method section the authors need to clarify the selected timeframe. Why the studied the period 2019-2024 and not a larger timeframe?
Authors response.
This cut-off point was considered as a criterion to determine the efforts derived from the changes in the approach that WADA under the International Standard Education presented in 2019. A brief statement explaining this determination was added in lines 176-182. In addition, this cut-off criteria were mentioned as limitation on the corresponding section.
Reviewer 1.
In the discussion section, again I think that the cognitive-affective distinction should be avoided.
Also, I think that the discussion would benefit from a more thorough interpretation of the findings. As it is, it largely reiterates the results of the review. I think a more thorough discussion on the what worked well and why or what didn’t worked well and why would be really meaningful and would add to visibility of the article. In addition, which variables were affected more by the interventions, which ones of them have been found more crucial in defining doping behavior etc, are important questions that the review could provide meaningful insights to anti-doping research.
Authors response
Thank you for the suggestions. According to the corrections made on the introduction section, as well as the ampliation of the analysis of the information throughout a meta-analysis, the discussion section was developed into the explanation of these new results.
Reviewer 1.
In line with the above, I think a section with practical implications derived from the review would be useful to the readers.
Authors response
Thank you again for your observation. A practical implication section was added where the main findings were explained to the potential practice for professionals and researchers
Reviewer 2 Report
Comments and Suggestions for Authors
Major Concerns
The manuscript states that 180 studies were identified, yet only 10 were included in the final review.
No clear justification is provided for why 170 studies were excluded. The absence of a well-defined screening process raises concerns about selection bias.
A systematic review should strive for comprehensiveness, yet the study window (2019–2024) is arbitrarily restrictive. Given that doping prevention strategies have evolved significantly over time, limiting the review to the past five years is unjustified.
There is no assessment of publication bias, which is particularly important in systematic reviews where negative or null results may be underreported.
The manuscript lacks a meaningful synthesis of findings. Instead, it presents a descriptive summary of individual studies without drawing clear comparative insights.
There is no discussion of effect sizes, confidence intervals, or statistical robustness, making it impossible to assess which interventions are most effective.
The paper does not distinguish between high-quality, well-controlled studies and weaker quasi-experimental designs, treating them as equally valid.
The majority of the included studies assess short-term knowledge gains, but none provide evidence of long-term behavioral change.
Without long-term follow-up data, the review cannot support claims that these interventions lead to sustained doping prevention.
The feasibility and scalability of these programs in real-world settings is not addressed. Practical challenges (e.g., cost, accessibility, compliance) are completely overlooked.
The study claims to use the JBI risk-of-bias tools, but there is no detailed reporting on how each study was scored.
Some included studies lack control groups, yet the authors fail to critically discuss how this impacts validity.
The manuscript does not differentiate between high-risk and low-risk studies, treating all included articles as equally reliable.
Many sections repeat the same findings in different ways without adding new insights.
The results section reads like an extended list of studies, rather than an integrated discussion of key themes.
The manuscript lacks clear transitions between sections, making it difficult to follow the logical flow.
Several citations are outdated or redundant and should be replaced with more recent systematic reviews or meta-analyses.
The abstract overstates the findings, suggesting strong evidence when in reality, the included studies vary significantly in quality.
The figures and tables are not optimally formatted, making data presentation cumbersome.
Final Recommendation: Major Revisions Required (or Reject and Resubmit)
Given the serious methodological limitations, lack of analytical depth, and weak real-world applicability, this manuscript cannot be accepted in its current form. The authors must substantially revise the study selection process, provide deeper statistical analysis, and improve the overall clarity and structure of the manuscript.
If these issues cannot be addressed, then a rejection would be warranted, as the current manuscript does not meet the standards for publication in a rigorous scientific journal.
Author Response
Dear Journal Editor of Sports,
I would like to begin by expressing my gratitude for the opportunity to contribute to this manuscript for such a prestigious journal. I also want to extend my sincere appreciation to the three reviewers who evaluated the manuscript—their commitment and expertise have undoubtedly played a crucial role in enhancing its quality.
Below, I provide a detailed response to each of the reviewers' observations, comments, and suggestions. I outline the corresponding explanations, the revisions made, and the decisions taken to improve the manuscript based on their valuable feedback.
Reviewer 2.
No clear justification is provided for why 170 studies were excluded. The absence of a well-defined screening process raises concerns about selection bias.
A systematic review should strive for comprehensiveness, yet the study window (2019–2024) is arbitrarily restrictive. Given that doping prevention strategies have evolved significantly over time, limiting the review to the past five years is unjustified.
Authors response
Thank you very much for the observation. The description of the article selection process was detailed in the methodology, the inclusion and exclusion criteria leading to the selection of these 10 articles are presented. Even, one article was retaken from the references of one reviewed article (it was also explained in the method section). Only publications that had conducted some kind of educational intervention for the prevention of doping were included. Experimental studies that did not employ an educational process were discarded from the analysis. No observational articles were included.
On the other hand, the cut-off point was considered as a criterion to determine the efforts derived from the efforts to orient the approach of the anti-doping educational intervention programs under the WADA International Standard Education presented in 2019. A brief statement explaining this determination was added in lines 176-182. In addition, these cut-off criteria were mentioned as limitations on the corresponding section.
Reviewer 2.
There is no assessment of publication bias, which is particularly important in systematic reviews where negative or null results may be underreported.
The study claims to use the JBI risk-of-bias tools, but there is no detailed reporting on how each study was scored.
The paper does not distinguish between high-quality, well-controlled studies and weaker quasi-experimental designs, treating them as equally valid.
Authors response
Thank you for your observation. We add the details of the assessment of study biases were aggregated in Table 1 and 2. Also, a note for the possible differences in the quality of the studies was made on the discussion section in the order to take care of the interpretation of the results from these studies.
Reviewer 2.
The manuscript lacks a meaningful synthesis of findings. Instead, it presents a descriptive summary of individual studies without drawing clear comparative insights.
There is no discussion of effect sizes, confidence intervals, or statistical robustness, making it impossible to assess which interventions are most effective.
The majority of the included studies assess short-term knowledge gains, but none provide evidence of long-term behavioral change.
Without long-term follow-up data, the review cannot support claims that these interventions lead to sustained doping prevention.
The feasibility and scalability of these programs in real-world settings is not addressed. Practical challenges (e.g., cost, accessibility, compliance) are completely overlooked.
Some included studies lack control groups, yet the authors fail to critically discuss how this impacts validity.
The manuscript does not differentiate between high-risk and low-risk studies, treating all included articles as equally reliable.
Many sections repeat the same findings in different ways without adding new insights.
The results section reads like an extended list of studies, rather than an integrated discussion of key themes.
Authors response
The authors really thank for your time, knowledge and comments to push us to really improve our manuscript. Analyses of effect size, confidence intervals and statistical robustness among the studies analyzed were conducted to conduct a meta-analysis of the data and explain the effectiveness of the programs. The results section was restructured to avoid redundancy and provide space for a more detailed analysis of the benefits and impacts of the educational interventions. The long-term effects of the reported studies were analyzed. The reliability and feasibility of intervention programs in real scenarios was analyzed, promoting an analysis of the effectiveness not only in terms of effects but also in terms of the cost-benefit of their implementation.
Reviewer 2.
The abstract overstates the findings, suggesting strong evidence when in reality, the included studies vary significantly in quality.
The figures and tables are not optimally formatted, making data presentation cumbersome.
Authors response
The wording of the abstract was modified and the Tables and figures were adjusted to the appropriate format.
Reviewer 3 Report
Comments and Suggestions for Authors
The authors need to address the following comments to be considered for its publication.
The introduction does not clearly state what existing systematic reviews lack, which makes it unclear why this review adds value. While socio-affective theories are mentioned, their direct relevance to doping prevention is underexplored, and the Theory of Planned Behaviour is briefly noted but not integrated into the study.
In lines 106-107, the authors mentioned that affective interventions specifically aim to strengthen intrinsic motivations, but this notion is unclear and needs to be strengthened.
The contents of active and passive participation can be inferred from SDT's three basic need theory, and the authors may strengthen the contents with SDT.
Only four databases (PubMed, ScienceDirect, Scielo, Redalyc) were searched. It is possible that omitting major sports science and psychology databases (e.g., Web of Science, PsycINFO, SportDiscus) may have introduced bias, potentially leading to a biased sample. The authors need to explain why they selected the four databases and why they selected publications after 2019.
The review encompasses both RCTs and quasi-experimental studies; however, there is a lack of clarity regarding how differences in study design were addressed in the analysis.
The discussion lacks insight into contextual differences (e.g., elite vs. amateur athletes, individual vs. team sports) that might affect intervention success.
The authros adrresed some theories in the Introduction, but they are not revisited in the Discussion.
The manuscript presents results in a narrative style, without reporting standardized effect sizes (e.g., Cohen's d, odds ratios). This makes it difficult to compare the effectiveness of different intervention approaches.
Several studies show no significant effect of interventions on doping susceptibility or moral disengagement. The reasons for this are unclear, and it is not known whether there were issues with study design, participant adherence, or sample size.
Given the systematic review nature of this study, a meta-analysis (or at least a forest plot) should be included to statistically summarise intervention effects.
The study claims that cognitive-affective interventions are superior, but there is no clear statistical evidence to support this. The question therefore arises as to whether there are studies in which cognitive or affective interventions alone have been equally effective. Furthermore, when selecting an intervention program, cost-effectiveness must be taken into account.
The discussion fails to incorporate contextual differences (e.g., elite vs. amateur athletes, individual vs. team sports) that may influence intervention efficacy.
While the authors have addressed certain theories in the Introduction, these theories are not revisited in the Discussion.
Author Response
Dear Journal Editor of Sports,
I would like to begin by expressing my gratitude for the opportunity to contribute to this manuscript for such a prestigious journal. I also want to extend my sincere appreciation to the three reviewers who evaluated the manuscript—their commitment and expertise have undoubtedly played a crucial role in enhancing its quality.
Below, I provide a detailed response to each of the reviewers' observations, comments, and suggestions. I outline the corresponding explanations, the revisions made, and the decisions taken to improve the manuscript based on their valuable feedback.
Reviewer 3.
The introduction does not clearly state what existing systematic reviews lack, which makes it unclear why this review adds value. While socio-affective theories are mentioned, their direct relevance to doping prevention is underexplored, and the Theory of Planned Behaviour is briefly noted but not integrated into the study.
In lines 106-107, the authors mentioned that affective interventions specifically aim to strengthen intrinsic motivations, but this notion is unclear and needs to be strengthened.
The contents of active and passive participation can be inferred from SDT's three basic need theory, and the authors may strengthen the contents with SDT.
Authors response
Thank you very much for your observation. Changes were made to the introduction, pointing out the gaps in existing literature reviews, as well as a more precise explanation of the relevance of theories of planned behavior and their impact on doping prevention and educational programs, as well as psychosocial theories related to moral and social aspects. Also, a more detailed explanation about the SDT influence on the athletes’ role participation relevance was added. You can see these changes in 42-47, 50-55, 77-86 lines.
Reviewer 3.
Only four databases (PubMed, ScienceDirect, Scielo, Redalyc) were searched. It is possible that omitting major sports science and psychology databases (e.g., Web of Science, PsycINFO, SportDiscus) may have introduced bias, potentially leading to a biased sample. The authors need to explain why they selected the four databases and why they selected publications after 2019.
Authors response
The use of these databases was determined due to the limited access by researchers to other databases such as web of science, psycInfo or sportdiscus, limiting us to open access databases in the world and Latin America. On the other hand, the cut-off point was considered as a criterion to determine the efforts derived from the efforts to orient the approach of the anti-doping educational intervention programs under the WADA International Standard Education presented in 2019. A brief statement explaining this determination was added in lines 176-182. In addition, these cut-off criteria were mentioned as limitations on the corresponding section.
Reviewer 3.
The review encompasses both RCTs and quasi-experimental studies; however, there is a lack of clarity regarding how differences in study design were addressed in the analysis.
Authors response
Thank you for your comment. Assessment of potential differences and methodological rigor of the studies was performed based on the JBI criteria. Details of the assessment of study biases are showed in Table 1 and 2. A timely analysis of how these data affect or may present differences in the results is added in the results and discussion.
Reviewer 3.
The manuscript presents results in a narrative style, without reporting standardized effect sizes (e.g., Cohen's d, odds ratios). This makes it difficult to compare the effectiveness of different intervention approaches.
Given the systematic review nature of this study, a meta-analysis (or at least a forest plot) should be included to statistically summarise intervention effects.
Several studies show no significant effect of interventions on doping susceptibility or moral disengagement. The reasons for this are unclear, and it is not known whether there were issues with study design, participant adherence, or sample size.
The study claims that cognitive-affective interventions are superior, but there is no clear statistical evidence to support this. The question therefore arises as to whether there are studies in which cognitive or affective interventions alone have been equally effective. Furthermore, when selecting an intervention program, cost-effectiveness must be taken into account.
Authors response
Thanks for this relevant comment and suggestion. We performed an analyses of effect size, confidence intervals and statistical robustness among the studies analyzed to conduct a meta-analysis of the data and explain the effectiveness of the programs. The results section was restructured to avoid redundancy and provide space for a more detailed analysis of the benefits and impacts of the educational interventions. The long-term effects of the reported studies were analyzed. The reliability and feasibility of intervention programs in real scenarios was analyzed, promoting an analysis of the effectiveness not only in terms of effects but also in terms of the cost-benefit of their implementation.
Reviewer 3.
The discussion lacks insight into contextual differences (e.g., elite vs. amateur athletes, individual vs. team sports) that might affect intervention success.
The authors addressed some theories in the Introduction, but they are not revisited in the Discussion.
The discussion fails to incorporate contextual differences (e.g., elite vs. amateur athletes, individual vs. team sports) that may influence intervention efficacy.
While the authors have addressed certain theories in the Introduction, these theories are not revisited in the Discussion.
Authors response
Thank you for the observations. The discussion was restructured by analyzing not only the distinctions between intervention approaches and strategies, but also by elaborating on the aspects that could have the greatest impact and explaining why this was the case... In addition, the variables that showed the greatest change and those that did not were detailed and explained. Furthermore, a paragraph was developed with practical implications for future interventions in doping prevention.
Round 2
Reviewer 2 Report
Comments and Suggestions for Authors
this version is better but please follow this recommendations :
- Methodological Observations
- Database Selection : The authors searched PubMed, ScienceDirect, SciELO, and Redalyc, which is commendable for including both international and Spanish-language databases. However, some databases widely used in sports science and psychology research—such as Scopus, Web of Science, and SPORTDiscus—were not mentioned. The absence of these could limit the overall comprehensiveness of the search.
- Time Window (2019–2024) : The rationale for including only studies from 2019 onward is understandable if the authors wanted to capture interventions in line with the newest WADA-ISE guidelines. Still, interventions published just prior to 2019 may well be informative. A brief explanation for this cutoff (beyond citing the 2019 WADA standards) may strengthen the justification for potentially overlooking older yet relevant studies.
- Heterogeneity of Included Studies ; The meta-analysis reports high I² values for some outcomes, reflecting substantial heterogeneity. Given that these interventions differ in design, duration, and context, it might be helpful to discuss in more detail how heterogeneity was handled, or to conduct sensitivity analyses that explore the impact of study design (RCT vs. quasi-experimental) or participant age on effect sizes.
- Moderators in the Meta-Regression: The paper highlights “active vs. passive” participation and “cognitive vs. affective” as primary moderators. While these are crucial dimensions, some additional potential moderators—such as age, competition level, or length of intervention—are also worth exploring or at least mentioning as directions for future research. Age brackets (e.g., adolescent vs. collegiate vs. elite-level athletes) might, for instance, experience and respond to doping-prevention education differently.
- Results and Discussion
- Emphasis on Short-Term vs. Long-Term Effects: A valuable insight from the results is that doping intentions and behaviors appear to improve in the short term but revert or weaken over time. The Discussion could benefit from more elaboration on possible mechanisms behind that drop-off. For instance, do athletes require booster sessions, mentorship, or “nudges” to maintain gains?
- Moral Disengagement and Values-Based Education: The findings that moral factors did not significantly improve—compared to doping intentions, for example—are notable. The Discussion does acknowledge that moral behavior is more challenging to shift with isolated interventions. Additional commentary on how future programs can reinforce moral values (perhaps by focusing on team culture or long-term mentoring) might give the reader concrete strategies or suggestions.
- Active vs. Passive Participation : The paper finds negative impacts when the athlete’s role is largely passive, underscoring the effectiveness of interactive, participatory interventions. This is an important take-home. The authors might consider giving examples of practical ways to embed active participation in real-world programs (e.g., role-playing, group discussions, or peer mentoring).
- Integration with Theoretical Models : The manuscript references cognitive, affective, and Self-Determination Theory approaches. This theoretical framing is strong. Still, further linking of specific results to these frameworks in the Discussion would clarify “why” certain approaches or roles might have a bigger impact. For instance, the reasons behind short-lived changes could be connected more closely to concepts like intrinsic versus extrinsic motivation.
The Conclusion does list implications, but reinforcing real-life takeaways in a separate “Practical Implications” subsection (or a set of bullet points at the end) could make the paper more impactful to coaches, sports administrators, and NADO program designers. For example: “Provide multiple sessions (instead of one-off interventions).” “Incorporate active and collaborative learning rather than purely didactic presentations.” “Emphasize motivational enhancement to avoid the relapse of doping behaviors after the program concludes.”
Author Response
Dear Reviewer,
We would like to express our sincere gratitude for your valuable time and insights, which have greatly contributed to improving our manuscript. Below, you will find our response to the comments and suggestions, who provided additional feedback on the second version of the manuscript. Additionally, we have highlighted the changes made to the document in red.
Reviewer 2.
- Methodological Observations
- Database Selection : The authors searched PubMed, ScienceDirect, SciELO, and Redalyc, which is commendable for including both international and Spanish-language databases. However, some databases widely used in sports science and psychology research—such as Scopus, Web of Science, and SPORTDiscus—were not mentioned. The absence of these could limit the overall comprehensiveness of the search.
Response.
Thank you for your observation. We acknowledge the significant limitation that the absence of high-prestige databases may pose to the quality of our manuscript. Unfortunately, we were unable to address this issue. To acknowledge this potential bias, we have provided a thorough explanation of this limitation in the limitations section of the manuscript.
Reviewer 2.
- Time Window (2019–2024) : The rationale for including only studies from 2019 onward is understandable if the authors wanted to capture interventions in line with the newest WADA-ISE guidelines. Still, interventions published just prior to 2019 may well be informative. A brief explanation for this cutoff (beyond citing the 2019 WADA standards) may strengthen the justification for potentially overlooking older yet relevant studies.
Response.
We appreciate the reviewer’s observation regarding the selected time frame (2019-2024) and the opportunity to clarify our decision. While we acknowledge that studies published before 2019 may contain valuable information, we established this time limit for two main reasons.
First, the World Anti-Doping Agency (WADA) guidelines have evolved significantly in recent years, and the 2019 updates reflect a more structured approach to anti-doping education (WADA, 2019). By focusing on studies published since 2019, we ensure that the interventions analyzed align with the latest standards and educational strategies.
Second, advancements in technology and pedagogy within anti-doping education have influenced how interventions are designed and implemented. Including only recent studies allows us to capture contemporary approaches that integrate new methodologies, such as interactive digital platforms and personalized learning strategies (Backhouse et al., 2021).
Nonetheless, we understand the reviewer’s concern and have incorporated a brief discussion in the manuscript to acknowledge the potential relevance of studies published before 2019, while further clarifying our rationale for this time frame selection.
Reviewer 2.
- Heterogeneity of Included Studies ; The meta-analysis reports high I² values for some outcomes, reflecting substantial heterogeneity. Given that these interventions differ in design, duration, and context, it might be helpful to discuss in more detail how heterogeneity was handled, or to conduct sensitivity analyses that explore the impact of study design (RCT vs. quasi-experimental) or participant age on effect sizes.
Response.
Thank you for your insightful comment. We acknowledge the high I² values observed in the meta-analysis, indicating substantial heterogeneity across the included studies. As you pointed out, the variations in intervention design, duration, and context could contribute to this heterogeneity. We explain the use of random-effects models, which allow for variability across studies, as well as the considerations made regarding the study characteristics.
Additionally, in response to your suggestion, we have conducted sensitivity analyses to explore the impact of key factors, such as study design (RCT vs. quasi-experimental) and participant age, and the length of intervention on the effect sizes. These analyses provide a deeper understanding of how different intervention features influence the outcomes and help clarify the robustness of the results. The findings from these sensitivity analyses will be included in the revised manuscript to better illustrate how heterogeneity was managed and to ensure the validity of the conclusions.
Reviewer 2.
- Moderators in the Meta-Regression: The paper highlights “active vs. passive” participation and “cognitive vs. affective” as primary moderators. While these are crucial dimensions, some additional potential moderators—such as age, competition level, or length of intervention—are also worth exploring or at least mentioning as directions for future research. Age brackets (e.g., adolescent vs. collegiate vs. elite-level athletes) might, for instance, experience and respond to doping-prevention education differently.
Response.
Thank you again for your relevant comments who push us to really improve our manuscript. We include the age and length of intervention as moderators for the analysis. The results are explained in the results section as well as discussed in the discussion section.
Reviewer 2.
- Results and Discussion
- Emphasis on Short-Term vs. Long-Term Effects: A valuable insight from the results is that doping intentions and behaviors appear to improve in the short term but revert or weaken over time. The Discussion could benefit from more elaboration on possible mechanisms behind that drop-off. For instance, do athletes require booster sessions, mentorship, or “nudges” to maintain gains?
- Moral Disengagement and Values-Based Education: The findings that moral factors did not significantly improve—compared to doping intentions, for example—are notable. The Discussion does acknowledge that moral behavior is more challenging to shift with isolated interventions. Additional commentary on how future programs can reinforce moral values (perhaps by focusing on team culture or long-term mentoring) might give the reader concrete strategies or suggestions.
- Active vs. Passive Participation : The paper finds negative impacts when the athlete’s role is largely passive, underscoring the effectiveness of interactive, participatory interventions. This is an important take-home. The authors might consider giving examples of practical ways to embed active participation in real-world programs (e.g., role-playing, group discussions, or peer mentoring).
- Integration with Theoretical Models : The manuscript references cognitive, affective, and Self-Determination Theory approaches. This theoretical framing is strong. Still, further linking of specific results to these frameworks in the Discussion would clarify “why” certain approaches or roles might have a bigger impact. For instance, the reasons behind short-lived changes could be connected more closely to concepts like intrinsic versus extrinsic motivation.
Response.
These comments were very important for the improvement of manuscript. We tried to response in the discussion section to every observation. In red color you can read the explanation for each comment, such the intervention effect on time, examples for moral and values-based interventions as well as how to implement an athletes’ active role during education programs and how theorical frameworks helps to explain the results of the meta-analysis.
Reviewer 2.
The Conclusion does list implications, but reinforcing real-life takeaways in a separate “Practical Implications” subsection (or a set of bullet points at the end) could make the paper more impactful to coaches, sports administrators, and NADO program designers. For example: “Provide multiple sessions (instead of one-off interventions).” “Incorporate active and collaborative learning rather than purely didactic presentations.” “Emphasize motivational enhancement to avoid the relapse of doping behaviors after the program concludes.”
Response.
Thank you very much for your insightful idea. We add bullet points as summarize of practical implication section, which we are sure help the impact of the manuscript.
Once again, on behalf of the authors, I would like to express our deep appreciation for the review process.
Sincerely,
Reviewer 3 Report
Comments and Suggestions for Authors
The authors have addressed the comments, and the manuscript has improved sufficiently for its publication. Thank the authors for their efforts on this work.
Author Response
Dear Reviewer,
On behalf of the authors, I would like to express our deep appreciation and gratitude for your valuable time and insights, which have greatly contributed to improving our manuscript.
Sincerely,